# DISCOUNTED ONLINE CONVEX OPTIMIZATION: UNIFORM REGRET ACROSS A CONTINUOUS INTERVAL

**Wenhao Yang**[1,2], **Sifan Yang**[1,2], **Lijun Zhang**[1,2,∗]
[1]National Key Laboratory for Novel Software Technology, Nanjing University, Nanjing, China
[2]School of Artificial Intelligence, Nanjing University, Nanjing, China
{yangwh, yangsf, zhanglj}@lamda.nju.edu.cn

## ABSTRACT

Reflecting the greater significance of recent history over the distant past in non-stationary environments, $\lambda$-discounted regret has been introduced in online convex optimization (OCO) to gracefully forget past data as new information arrives. When the discount factor $\lambda$ is given, online gradient descent with an appropriate step size achieves an $O(1/\sqrt{1-\lambda})$ discounted regret. However, the value of $\lambda$ is often not predetermined in real-world scenarios. This gives rise to a significant *open question*: is it possible to develop a discounted algorithm that adapts to an unknown discount factor. In this paper, we affirmatively answer this question by providing a novel analysis to demonstrate that smoothed OGD (SOGD) achieves a uniform $O(\sqrt{\log T/1-\lambda})$ discounted regret, holding for all values of $\lambda$ across a continuous interval simultaneously. The basic idea is to maintain multiple OGD instances to handle different discount factors, and aggregate their outputs sequentially by an online prediction algorithm named as Discounted-Normal-Predictor (DNP). Our analysis reveals that DNP can combine the decisions of two experts, even when they operate on discounted regret with different factors.

## 1 INTRODUCTION

Online convex optimization (OCO) serves as a fundamental framework for online learning, effectively modeling a wide range of real-world sequential prediction and decision-making problems (Hazan, 2016). OCO can be viewed as a repeated game between the learner and the environment, governed by the following protocol. In each round $t \in [T]$, the learner chooses a decision $\mathbf{w}_t$ from a convex domain $\mathcal{W} \subseteq \mathbb{R}^d$. Then, the learner suffers a loss $f_t(\mathbf{w}_t)$ and observe some information about the functions, where $f_t: \mathcal{W} \mapsto \mathbb{R}$ is chosen by the environment. To evaluate the performance of the learner, static regret is commonly used (Cesa-Bianchi & Lugosi, 2006):

$$\text{Regret}(T) = \sum_{t=1}^{T} f_t(\mathbf{w}_t) - \min_{\mathbf{w} \in \mathcal{W}} \sum_{t=1}^{T} f_t(\mathbf{w})$$

which is defined as the difference between the cumulative loss of the online learner and that of the best decision chosen in hindsight. However, static regret is not well-suited for changing environments where the future significantly diverges from the past. To facilitate gracefully forgetting past data as new information arrives, $\lambda$-discounted regret has been proposed (Freund & Hsu, 2008):

$$\text{D-Regret}(T, \lambda) = \sum_{t=1}^{T} \lambda^{T-t} f_t(\mathbf{w}_t) - \min_{\mathbf{w} \in \mathcal{W}} \sum_{t=1}^{T} \lambda^{T-t} f_t(\mathbf{w}) \tag{1}$$

where $\lambda \in (0, 1)$ is the discount factor, denoting the degree of forgetting of the past.

Although discounted regret has been explored to some extent in prediction with expert advice (PEA) (Freund & Hsu, 2008; Chernov & Zhdanov, 2010; Kapralov & Panigrahy, 2010; Cesa-bianchi et al., 2012; Krichene et al., 2014) and games (Brown & Sandholm, 2019; Xu et al., 2024), discounted OCO has been relatively underexplored in the literature. Recently, Zhang et al. (2024) studied

---

[∗]Lijun Zhang is the corresponding author.

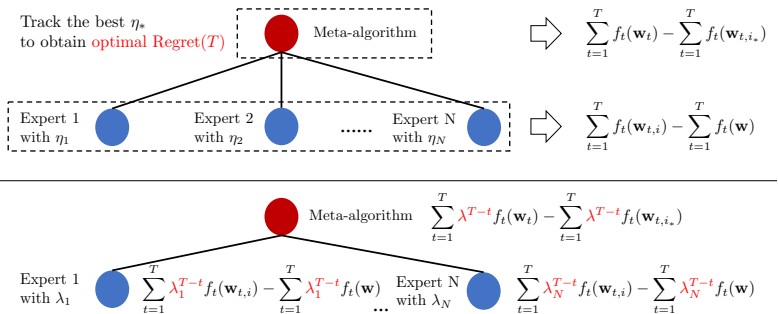

Figure 1: A meta-expert framework for OCO that adapts to unknown parameters (previous work, upper panel) and discounted OCO (our setting, lower panel). Previous work typically decomposes the regret into the sum of meta-regret and expert-regret, where $\mathbf{w}_{t,i}$ denotes the decision of the $i$-th expert. However, this method fails to combine these experts' decisions under discounted OCO.

discounted OCO with given discount factors, aiming to establish gradient-adaptive regret bounds. For a fixed factor, they derive an $O(1/\sqrt{1-\lambda^2})$ result for $\lambda$-discounted regret, but their algorithm requires the prior knowledge of discount factor $\lambda$. However, in many decision-making problems, the discount factor reflects intrinsic dynamics rather than a tunable hyperparameter. For instance, in intertemporal economic models, the discount factor represents agents' preferences, which are shaped by empirical observations of market behavior and validated through economic studies (Cohen et al., 2020). Crucially, in such models the discount factor is treated as a genuine parameter grounded in economic reality rather than an arbitrary design choice. This highlights the existence of a "true" discount factor that inherently resides in actual contexts, rather than freely specified in advance. Recognizing the considerable significance of adapting to an unknown discount factor in practical scenarios, Zhang et al. (2024) explicitly leave this as an important *open question* in their work.

Notably, there exists a similar performance measurement for dealing with changing environments, known as adaptive regret. Formally, the adaptive regret is defined as the maximal static regret for every interval $[r, s] \subseteq [T]$ over the whole time horizon:

$$\text{SA-Regret}(T, \tau) = \max_{[s, s+\tau-1] \subseteq [T]} \left\{ \sum_{t=s}^{s+\tau-1} f_t(\mathbf{w}_t) - \min_{\mathbf{w} \in \mathcal{W}} \sum_{t=s}^{s+\tau-1} f_t(\mathbf{w}) \right\}$$

where $\tau$ is the interval length. It is evident that adaptive regret serves a similar purpose to that of discounted regret, as both effectively define a temporal horizon of interest. Specifically, the length of an interval in adaptive regret is conceptually analogous to the effective window size controlled by a discount factor; for instance, a smaller discount factor emphasizes recent data by promoting more rapid forgetting of the past, which corresponds to a focus on more recent intervals. The best-known algorithm (Jun et al., 2017) achieves $O(\sqrt{\tau \log T})$ strongly adaptive regret for all interval length. Compared to the minimax optimal result for static regret, their bound incurs an additional $\log T$ term, which is the necessary cost of adaptivity across all intervals. Since existing algorithms for adaptive regret (Hazan & Seshadhri, 2007; Jun et al., 2017; Zhang et al., 2018b; Wan et al., 2022; Wang et al., 2024; Yang et al., 2024) provide guarantees that hold simultaneously for all intervals, it is natural to ask *whether it is possible to design a discounted OCO algorithm that adapts to an unknown discount factor*. In this paper, we provide an affirmative answer.

## 1.1 TECHNICAL CHALLENGE

In the literature, extensive research has explored online algorithms that adapt to unknown parameters, including universal OCO (van Erven & Koolen, 2016; Zhang et al., 2022b; 2025), dynamic regret (Zhang et al., 2018a; Baby & Wang, 2021; Zhao et al., 2024), and adaptive regret (Daniely et al., 2015; Jun et al., 2017). These methods adopt the meta-expert framework as shown in the upper panel of Figure 1, where they maintain multiple experts with different configurations and deploy a meta-algorithm to track the best one. Therefore, to adapt to an unknown discount factor, a straightforward idea is to apply this meta-expert framework by constructing multiple OGD instances, each operating with a different potential discount factor, and then using a meta-algorithm to combine their decisions.

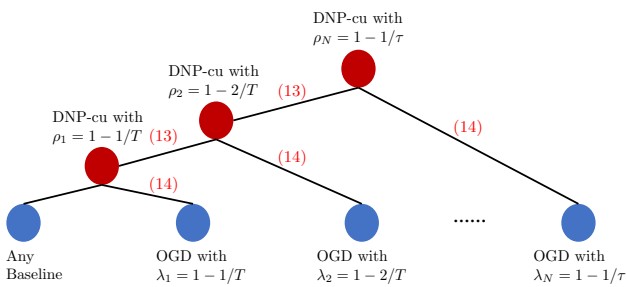

Figure 2: Overall procedure of our method: sequentially aggregation by DNP-cu with different discount factors (red nodes) of OGD experts (blue nodes), using meta-regret from (13) and (14).

However, this strategy fails to handle the discounted scenario, because existing meta-algorithms used in the these studies, such as Hedge (Freund & Schapire, 1997) or Fixed-Share (Herbster & Warmuth, 1998), typically require that all experts and the meta-algorithm operate under *a unified performance measurement*. For discounted OCO, the fact that the uncertainty of the discount factor is tied to the performance measurement makes this requirement difficult to satisfy. As depicted in the lower panel of Figure 1, experts configured for different discount factors are essentially operating under *different performance measures*, i.e., $\lambda$-discounted regret with varying $\lambda$. This renders the traditional meta-expert framework incapable of effectively handling an unknown discount factor.

### 1.2 OUR SOLUTION AND CONTRIBUTIONS

To address the above challenge, we revisit Smoothed OGD (SOGD) (Zhang et al., 2022a), which is proposed to support the adaptive regret for smoothed OCO. The key idea of their work is to construct multiple instances of OGD with different step sizes, and employ Discounted-Normal-Predictor with conservative updating (DNP-cu) (Kapralov & Panigrahy, 2010; 2011) as the meta-algorithm to sequentially aggregate their decisions. Following this idea, we first discretize the continuous interval of potential discount factors, say $\lambda \in [1 - 1/\tau, 1 - 1/T]$ where $\tau$ is a minimal window length, by constructing a geometric series to cover the range of their values. For each possible discount factor $\lambda_i$, we create an expert by running an instance of OGD to achieve optimal $\lambda_i$-discounted regret. Then, we employ multiple instances of DNP-cu with different discount factors $\rho_i = \lambda_i$ to sequentially aggregate decisions from each expert. The overall procedure is illustrated in Figure 2. In this work, we analyze the performance of DNP-cu under the discounted payoff setting, demonstrating its ability to effectively control the discounted regret. Furthermore, our novel analysis reveals that DNP-cu is able to successfully aggregate the decisions of two experts, *even when they operate on discounted regret with different discount factors*. Significantly, we prove that this approach achieves a uniform $O(\sqrt{\log T / 1 - \lambda})$ bound for $\lambda$-discounted regret that holds simultaneously for all $\lambda \in [1 - 1/\tau, 1 - 1/T]$, and does not require knowing the value of $\lambda$.

Finally, we would like to emphasize that although the idea of deriving uniform discounted regret across a continuous interval has appeared in Kapralov & Panigrahy (2010), their analysis is conducted in the setting of PEA rather than OCO. Moreover, Kapralov & Panigrahy (2010, § 5) only outlined the proof sketch without providing many of the technical details. Reconstructing the complete argument and addressing the gaps and inaccuracies in their presentation requires significant effort. More discussion on our technical contribution can be found in Appendix C.

## 2 RELATED WORKS

In this section, we review related works to our paper, including discounted online learning, online convex optimization with memory and discounted-normal-predictor.

### 2.1 DISCOUNTED ONLINE LEARNING

Due to the fact that recent information is often more important than past history in non-stationary environments, discounted online learning has been proposed to gradually forget the past as new data

---

**Algorithm 1** Discounted-Normal-Predictor

---

**Require:** Two parameters: $\rho$ and $Z$
 1: Set $x_1 = 0$
 2: **for** $t = 1, \ldots, T$ **do**
 3:    Predict $g(x_t)$
 4:    Receive $b_t$
 5:    Set $x_{t+1} = \rho x_t + b_t$
 6: **end for**

---

arrives. Under the setting of PEA, discounted regret is defined as (Cesa-Bianchi & Lugosi, 2006):

$$\sum_{t=1}^{T} \beta_{T-t} \mathbf{p}_t^\top \boldsymbol{\ell}_t - \min_{\mathbf{p} \in \Delta_N} \sum_{t=1}^{T} \beta_{T-t} \mathbf{p}^\top \boldsymbol{\ell}_t, \tag{2}$$

where $\mathbf{p}_t$ is a weight picked from a simplex $\Delta_N$, $\{\beta_t\}_{t=1}^{T}$ is a decreasing sequence of discount factors and $N$ is the number of experts. The $\lambda$-discounted regret (1) studied in our work can be viewed as a special case of (2), also referred to as exponential discounting. The seminal work of Freund & Hsu (2008) proposes a discounted variant of Hedge, which achieves an $O(\sqrt{\ln N/(1-\lambda)})$ regret bound. Subsequent works (Chernov & Zhdanov, 2010; Cesa-bianchi et al., 2012; Krichene et al., 2014) have also explored other discounted variants under the framework of "tracking the best expert". Furthermore, Brown & Sandholm (2019); Xu et al. (2024) propose discounted version of counterfactual regret minimization (CFR) for solving imperfect-information games. Recently, discounted regret has gained attention in the context of OCO. Zhang et al. (2024) investigate adaptive OGD and FTRL for discounted regret with time-varying factors. For online linear regression, Jacobsen & Cutkosky (2024) present a discounted variant of the VAW forecaster, which enjoys dynamic regret guarantees. Furthermore, they also explore strongly adaptive regret. Based on online-to-non-convex conversion (Cutkosky et al., 2023), Ahn et al. (2024) investigate Adam optimizer, and propose a conversion from the discounted regret to the dynamic regret. This coversion is recently refined by Xie et al. (2026). However, these aforementioned works cannot adapt to an unknown discount factor.

## 2.2 ONLINE CONVEX OPTIMIZATION WITH MEMORY

Online convex optimization (OCO) with memory is a related line of work to discouted OCO. Anava et al. (2015) extend classical OCO to OCO with finite memory framework, that allows the current loss to depend on a constant number of past decisions. Subsequently, Kumar et al. (2023) introduces OCO with unbounded memory, which can capture the complete long-term dependence of current losses on past decisions and model complex temporal-structure applications (Agarwal et al., 2019). Specifically, to evaluate the performance of an algorithm $\mathcal{A}$, they define the policy regret as the measurement Arora et al. (2012):

$$\text{P-Regret}(T) = \sum_{t=1}^{T} f_t(h_t) - \min_{\mathbf{w} \in \mathcal{W}} \sum_{t=1}^{T} f_t \left( \sum_{k=0}^{t-1} A^k B \mathbf{w} \right),$$

where the history space $h_t \in \mathcal{H}$ is defined as $h_t = A h_{t-1} + B \mathbf{w}_t$, and $A$ and $B$ are linear operators. Their work considers a special case with $\rho$-discounted infinite memory, where the operators are defined as $A(y_0, y_1, \cdots) = (0, \rho y_0, \rho y_1, \cdots)$ and $B(x) = (x, 0, \cdots)$. In this manner, the history becomes $h_t = (\mathbf{w}_t, \rho \mathbf{w}_{t-1}, \rho^2 \mathbf{w}_{t-2}, \cdots)$, which places greater emphasis on recent actions while diminishing the influence of actions taken further in the past. Along this line of research, several follow-up works have investigated various problems under the setting of OCO with memory (Zhao et al., 2023; Shi et al., 2020; Lin et al., 2023).

The difference between discounted OCO and OCO with memory lies in the fundamentally different ways in which past losses are incorporated. In discounted OCO, the discount factor is applied directly to the final loss function and is tied to the performance measure. In contrast, OCO with memory integrates the entire history of decisions, and the extent of historical influence is determined by the size of the memory window.

## 2.3 DISCOUNTED-NORMAL-PREDICTOR

Discounted-Normal-Predictor (DNP) (Kapralov & Panigrahy, 2010) was introduced to solve the bit prediction problem, with its protocol described as follows. Consider an adversarial sequence of bits $b_1, \ldots, b_T$ where each $b_t \in [-1, 1]$ can take real values, and our goal is to predict the next bit from the previous bits. In each round $t \in [T]$, the learner predicts a confidence $c_t \in [-1, 1]$, and observes the value of $b_t$. Then, the learner gets a payoff $c_t b_t$ at each round. For a $T$-round game, the goal of the learner is to maximize the cumulative payoff $\sum_{t=1}^{T} c_t b_t$.

The overall procedure of DNP is summarized in Algorithm 1. In each round $t$, DNP maintains a discounted deviation defined as $x_t = \sum_{j=1}^{t-1} \rho^{t-1-j} b_j$, where the discount factor $\rho = 1 - 1/n$ and $n > 0$ is a parameter for the interval length. In Step 3, DNP predicts the confidence level $g(x_t)$ by a confidence function $g(\cdot)$, which is defined as

$$g(x) = \text{sign}(x) \cdot \min\left( Z \cdot \text{erf}\left( \frac{|x|}{4\sqrt{n}} \right) e^{\frac{x^2}{16n}}, 1 \right) \tag{3}$$

where $Z > 0$ is a parameter, and $\text{erf}(x) = \frac{2}{\sqrt{\pi}} \int_0^x e^{-s^2} ds$ is the error function. For any $Z \leq 1/e$, Kapralov & Panigrahy (2010, Theorem 14) have proved that DNP satisfies

$$\sum_{t=1}^{T} g(x_t) b_t \geq \max\left( \left| \sum_{j=1}^{T} b_j \right| - O\big(\sqrt{T \log(1/Z)}\big), -O\big(Z\sqrt{T}\big) \right) \tag{4}$$

where we set $n = T$ in Algorithm 1. Compared to the strategy that predicts the majority bit with payoff $|\sum_j b_j|$, DNP achieves an $O(\sqrt{T \log T})$ regret by setting $Z = o(1/T)$, as well as a subconstant $o(1)$ loss. Furthermore, DNP can combine the decisions of two experts by defining $b_t$ as the difference between the losses of experts and restricting $c_t \in [0, 1]$. For the general case of $N$ experts, we can use multiple DNP to aggregate experts' decisions one by one.

Kapralov & Panigrahy (2010, Lemma 18) also investigate the discounted payoff $\sum_{t=1}^{T} \rho^{T-t} c_t b_t$, and propose a conservative updating rule to control the value of the deviation $x_t$. To be specific, Line 5 of Algorithm 1 is replaced by

**if** $|x_t| < U(n)$ or $g(x_t) b_t < 0$ **then**
    Set $x_{t+1} = \rho x_t + b_t$
**else**
    Set $x_{t+1} = \rho x_t$
**end if**

where $U(n) = O(\sqrt{n \log(1/Z)})$ is a constant such that $g(x) = 1$ for $|x| \geq U(n)$. It can be seen that the current bit $b_t$ is utilized to update $x_t$ only when the confidence of the algorithm is low or the algorithm predicts incorrectly. Then, they demonstrate that the discounted payoff is on the order of $O(-Z/(1 - \rho))$ (Kapralov & Panigrahy, 2010, Lemma 18). Similarly, Discounted-Normal-Predictor with conservative updating can be applied to the problem of PEA.

Following their work, Daniely & Mansour (2019) refine the analysis of DNP to support the switching cost and the adaptive regret. They slightly modify the confidence function as

$$g(x) = \Pi_{[0,1]} [\tilde{g}(x)], \quad \tilde{g}(x) = \sqrt{\frac{n}{8}} Z \cdot \text{erf}\left( \frac{x}{\sqrt{8n}} \right) e^{\frac{x^2}{16n}} \tag{5}$$

where $\Pi_{[0,1]}[\cdot]$ denotes the projection operation onto the set $[0, 1]$, and the error function is redefined as $\text{erf}(x) = \int_0^x e^{-s^2/2} ds$. Daniely & Mansour (2019, Theorem 10) demonstrates that their refined DNP with projection operation attains similar regret bounds to that of (4) even in the presence of switching costs. Moreover, they also establish adaptive regret guarantee. Subsequently, Zhang et al. (2022a) analyze the performance of DNP with conservative updating (DNP-cu) in the context of OCO, and propose Smoothed OGD (SOGD) algorithm, where multiple instances of OGD with different step sizes are created and aggregated sequentially using DNP-cu. Their analysis shows that SOGD achieves nearly-optimal bounds for adaptive regret and dynamic regret, in the presence of switching cost. However, both of their studies do not consider discounted regret.

## 3 MAIN RESULTS

In this section, we introduce standard assumptions of OCO, followed by our discounted algorithms.

### 3.1 PRELIMINARIES

**Assumption 1.** *All the functions $f_t$'s are convex over the domain $\mathcal{W}$.*

**Assumption 2.** *The gradients of all functions are bounded by $G$, i.e.,*

$$\max_{\mathbf{w} \in \mathcal{W}} \|\nabla f_t(\mathbf{w})\| \leq G, \ \forall t \in [T]. \tag{6}$$

**Assumption 3.** *The diameter of the domain $\mathcal{W}$ is bounded by $D$, i.e.,*

$$\max_{\mathbf{w}, \mathbf{w}' \in \mathcal{W}} \|\mathbf{w} - \mathbf{w}'\| \leq D. \tag{7}$$

Without loss of generality, we assume (Zhang et al., 2022a)

$$f_t(\mathbf{w}) \in [0, GD], \ \forall \mathbf{w} \in \mathcal{W}, \ t \in [T]. \tag{8}$$

### 3.2 ONLINE GRADIENT DESCENT FOR DISCOUNTED OCO

We investigate online gradient descent (OGD) with constant step size (Zinkevich, 2003) for discounted OCO. OGD performs gradient descent to update the current solution $\mathbf{w}_t$:

$$\mathbf{w}_{t+1} = \Pi_{\mathcal{W}} \big[ \mathbf{w}_t - \eta \nabla f_t(\mathbf{w}_t) \big] \tag{9}$$

where $\eta > 0$ is the step size, and $\Pi_{\mathcal{W}}[\cdot]$ denotes the projection onto $\mathcal{W}$. In the following, we present the discounted regret of OGD.

**Theorem 1.** *Under Assumptions 1, 2 and 3, for any $\mathbf{w} \in \mathcal{W}$ and a given $\lambda \in (0, 1)$, OGD satisfies*

$$\sum_{t=1}^{T} \lambda^{T-t} f_t(\mathbf{w}_t) - \sum_{t=1}^{T} \lambda^{T-t} f_t(\mathbf{w}) \leq \frac{DG\sqrt{2}}{\sqrt{1-\lambda}}$$

*where we set $\eta = D\sqrt{2(1-\lambda)}/G$. Furthermore, in the (nearly)-static regime $\lambda \in [1 - 1/T, 1]$, one can instead set $\eta = D/(G\sqrt{T})$, which gives*

$$\sum_{t=1}^{T} \lambda^{T-t} f_t(\mathbf{w}_t) - \sum_{t=1}^{T} \lambda^{T-t} f_t(\mathbf{w}) \leq \frac{3}{2} DG\sqrt{T}.$$

**Remark:** The first part of Theorem 1 shows that OGD achieves an $O(1/\sqrt{1-\lambda})$ bound for $\lambda$-discounted regret, which is on the same order as $O(1/\sqrt{1-\lambda^2})$ established by Zhang et al. (2024, Theorem 6) because $1 - \lambda < 1 - \lambda^2 < 2(1-\lambda)$. It is worth mentioning that both the step size and the upper bound are independent of the total iterations $T$, thus the $O(1/\sqrt{1-\lambda})$ bound holds uniformly over time. Furthermore, when $\lambda \in [1 - 1/T, 1]$, the effective window size is comparable to or larger than the whole horizon, and the alternative choice $\eta = D/(G\sqrt{T})$ gives an $O(\sqrt{T})$ bound.

### 3.3 UNIFORM DISCOUNTED REGRET ACROSS A CONTINUOUS INTERVAL

In this subsection, we focus on the more challenging case with an unknown discount factor. We begin by discussing the range of $\lambda$ values that are of interest. With a discount factor $0 < \lambda < 1$, the effective window size is essentially $\frac{1}{1-\lambda}$. For a $T$-round game, it is natural to require

$$\frac{1}{1-\lambda} \in [\tau, T]$$

where $\tau$ is a minimal window length introduced for technical reasons, implying [1]

$$\lambda \in \left[ 1 - \frac{1}{\tau}, 1 - \frac{1}{T} \right]. \tag{10}$$

---

[1] While there is no particular reason to avoid setting the upper bound of $\lambda$ to $1 - 1/T^{\alpha}$ for some $\alpha \geq 1$, we focus on $1 - 1/T$ for simplicity.

---

**Algorithm 2** Discounted-Normal-Predictor with conservative updating (DNP-cu)

---

**Require:** Two parameters: $\rho$ and $Z$
1: Set $x_1 = 0$, $n = 1/(1 - \rho)$, and $U(n) = \tilde{g}^{-1}(1)$
2: **for** $t = 1, \ldots, T$ **do**
3:     Predict $g(x_t)$ where $g(\cdot)$ is defined in (5)
4:     Receive $b_t$
5:     **if** $x_t \in [0, U(n)]$ or $x_t < 0 \& b_t > 0$ or $x_t > U(n) \& b_t < 0$ **then**
6:         Set $x_{t+1} = \rho x_t + b_t$
7:     **else**
8:         Set $x_{t+1} = \rho x_t$
9:     **end if**
10: **end for**

---

As shown in Theorem 1, OGD with a suitable step size can minimize the discounted regret for a particular value of $\lambda$. In order to handle all possible values of $\lambda$ in (10), we discretize the interval $[1 - 1/\tau, 1 - 1/T]$ by introducing the following set:

$$\mathcal{S} = \left\{ 1 - \frac{1}{T}, 1 - \frac{2}{T}, \ldots, 1 - \frac{2^N}{T} \right\}, \text{ where } N = \left\lceil \log_2 \frac{T}{\tau} \right\rceil \tag{11}$$

which covers the range of discount factor values. Then, for each discount factor $\lambda_i = 1 - 2^{i-1}/T \in \mathcal{S}$, we create an instance of OGD, denoted by $\mathcal{A}_i$. According to Theorem 1, the step size of $\mathcal{A}_i$ is set as

$$\eta_i = \frac{D\sqrt{2(1 - \lambda_i)}}{G} = \frac{D}{G}\sqrt{\frac{2^i}{T}} \tag{12}$$

so that it achieves the optimal $\lambda_i$-discounted regret. As discussed in Section 1.1, the traditional meta-expert framework is unable to combine these experts' decisions to attain optimal $\lambda$-discounted regret with an unknown $\lambda$. To address this issue, we choose Discounted-Normal-Predictor with conservative updating (DNP-cu) (Kapralov & Panigrahy, 2010) (summarized in Algorithm 2) as the meta-algorithm to sequentially aggregate the decisions from these multiple experts.

Before describing the specific algorithm, we first present the performance of DNP-cu (i.e., Algorithm 2) under the discounted payoff setting. Although Zhang et al. (2022a, Theorem 1) have investigated the properties of DNP-cu, their analysis is restricted to the standard payoff.

**Theorem 2.** *Suppose $Z \leq \frac{1}{e}$, $n \geq \max\{8e, 16 \log \frac{1}{Z}\}$ and $U(n) \geq 22$. For any bit sequence $b_1, \ldots, b_T$ where $|b_t| \leq 1$, and any $\eta \geq \rho = 1 - 1/n$, the discounted payoff of Algorithm 2 satisfies*

$$\sum_{t=1}^{T} \eta^{T-t} g(x_t) b_t \geq -\frac{Z}{2(1 - \eta)}. \tag{13}$$

*Furthermore, it also satisfies*

$$\sum_{t=1}^{T} \rho^{T-t} g(x_t) b_t \geq \sum_{t=1}^{T} \rho^{T-t} b_t - \frac{Z}{2(1 - \rho)} - U(n) - 1 \tag{14}$$

*where*

$$U(n) = \tilde{g}^{-1}(1) \leq \sqrt{16n \log \frac{1}{Z}}. \tag{15}$$

**Remark:** Theorem 2 indicates that DNP-cu can effectively control the discounted payoff. First, (14) shows that DNP-cu is able to support the $\lambda$-discounted payoff when $\rho = \lambda$. Furthermore, (13) reveals that while DNP-cu operates with a discount factor $\rho$, it can also provide a discounted payoff guarantee for a different discount factor $\eta$, provided that $\eta \geq \rho$. Therefore, we can exploit (13) and (14) to enable aggregation of two experts operating under the discounted regret with different discount factors, as specified below.

Algorithm 3, referred to as Combiner, serves as a meta-algorithm to aggregate the outputs of two OGD experts. Let $\mathcal{A}_1$ and $\mathcal{A}_2$ denote two OGD algorithm, and let $\mathbf{w}_{t,1}$ and $\mathbf{w}_{t,2}$ be their respective predictions at round $t$. Combiner generates a convex combination of $\mathbf{w}_{t,1}$ and $\mathbf{w}_{t,2}$ as its output:

$$\mathbf{w}_t = (1 - \omega_t)\mathbf{w}_{t,1} + \omega_t \mathbf{w}_{t,2} \tag{16}$$

---

**Algorithm 3** Combiner

---

**Require:** Two parameters: $\rho$ and $Z$
**Require:** Two algorithms: $\mathcal{A}_1$ and $\mathcal{A}_2$
 1: Let $\mathcal{D}$ be an instance of DNP-cu, i.e., Algorithm 2, with parameter $\rho$ and $Z$
 2: Receive $\mathbf{w}_{1,1}$ and $\mathbf{w}_{1,2}$ from $\mathcal{A}_1$ and $\mathcal{A}_2$ respectively
 3: Receive the prediction $\omega_1$ from $\mathcal{D}$
 4: **for** $t = 1, \dots, T$ **do**
 5:     Predict $\mathbf{w}_t = (1 - \omega_t)\mathbf{w}_{t,1} + \omega_t\mathbf{w}_{t,2}$
 6:     Send the loss function $f_t(\cdot)$ to $\mathcal{A}_1$ and $\mathcal{A}_2$
 7:     Receive $\mathbf{w}_{t+1,1}$ and $\mathbf{w}_{t+1,2}$ from $\mathcal{A}_1$ and $\mathcal{A}_2$ respectively
 8:     Send the real bit $\ell_t = (f_t(\mathbf{w}_{t,1}) - f_t(\mathbf{w}_{t,2}))/GD$ to $\mathcal{D}$
 9:     Receive the prediction $\omega_{t+1}$ from $\mathcal{D}$
10: **end for**

---

where the weight $\omega_t \in [0, 1]$. By the convexity of $f_t(\cdot)$, we have

$$f_t(\mathbf{w}_t) \leq (1 - \omega_t)f_t(\mathbf{w}_{t,1}) + \omega_t f_t(\mathbf{w}_{t,2}). \tag{17}$$

Summing (17) over $\sum_{t=1}^{T} \lambda^{T-t}$ and rearranging the terms, the $\lambda_1, \lambda_2$-discounted regret of Combiner with respect to $\mathcal{A}_1$ and $\mathcal{A}_2$ can be bounded as follows:

$$\sum_{t=1}^{T} \lambda_1^{T-t} f_t(\mathbf{w}_t) - \sum_{t=1}^{T} \lambda_1^{T-t} f_t(\mathbf{w}_{t,1}) \leq - GD \sum_{t=1}^{T} \lambda_1^{T-t} \omega_t \ell_t, \tag{18}$$

$$\sum_{t=1}^{T} \lambda_2^{T-t} f_t(\mathbf{w}_t) - \sum_{t=1}^{T} \lambda_2^{T-t} f_t(\mathbf{w}_{t,2}) \leq - GD \sum_{t=1}^{T} \lambda_2^{T-t} (\omega_t \ell_t - \ell_t) \tag{19}$$

where we define

$$\ell_t = \frac{f_t(\mathbf{w}_{t,1}) - f_t(\mathbf{w}_{t,2})}{GD} \overset{(8)}{\in} [-1, 1]. \tag{20}$$

To determine the weight, we pass $\ell_t$ to DNP-cu and set $\omega_t$ as its output. By the theoretical guarantee of DNP-cu in Theorem 2, we can use (13) and (14) to upper bound the discounted regret (18) and (19), respectively. Notably, since the discounted payoff (13) in Theorem 2 supports arbitrary discount factor $\eta \geq \rho$, we are able to successfully aggregate the decisions of two experts for different discounted regret measurements. Specifically, let $\mathcal{A}_1$ and $\mathcal{A}_2$ be OGD algorithms for discount factors $\lambda_1$ and $\lambda_2$ respectively, with $\mathbf{w}_{t,1}$ and $\mathbf{w}_{t,2}$ as their decisions. To combine their decisions, we employ DNP-cu with $\rho = \lambda_2$ to obtain:

$$\sum_{t=1}^{T} \lambda_1^{T-t} f_t(\mathbf{w}_t) - \sum_{t=1}^{T} \lambda_1^{T-t} f_t(\mathbf{w}_{t,1}) \overset{(13),(18)}{\leq} \frac{GDZ}{2(1 - \lambda_1)},$$

$$\sum_{t=1}^{T} \lambda_2^{T-t} f_t(\mathbf{w}_t) - \sum_{t=1}^{T} \lambda_2^{T-t} f_t(\mathbf{w}_{t,2}) \overset{(14),(19)}{\leq} GD\left(\frac{Z}{2(1 - \lambda_2)} + U(n) + 1\right)$$

where $\mathbf{w}_t$ is the combined output of $\mathbf{w}_{t,1}$ and $\mathbf{w}_{t,2}$, and we require $\eta = \lambda_1 \geq \lambda_2$ by Theorem 2.

**Remark:** By setting $Z = 1/T$, DNP-cu delivers a small meta-regret with respect to the decision from $\mathcal{A}_1$ and $\mathcal{A}_2$. Combining the above inequalities with the corresponding $\lambda_i$-discounted regret achieved by the $i$-th expert, we demonstrate that when the discount factor $\lambda \in \{\lambda_1, \lambda_2\}$, our method can achieve the optimal $O(\sqrt{1/(1-\lambda)})$ bound for $\lambda$-discounted regret without requiring knowledge of the exact value of the discount factor.

To establish uniform discounted regret over the range of discount factors specified in (10), we construct multiple experts by running OGD with different step sizes in (11), and apply Combiner to sequentially aggregate these algorithms. For each $i \in [N + 1]$, we create an instance of Combiner, denoted by $\mathcal{B}_i$, to combine $\mathcal{B}_{i-1}$ and $\mathcal{A}_i$, where the discount factor of $\mathcal{B}_i$ is set to $\lambda_i$, matching that of $\mathcal{A}_i$. Initially, we set $\mathcal{B}_0$ as any baseline, such as an algorithm that predicts a fixed point in $\mathcal{W}$. At each round $t$, we sequentially run $\mathcal{B}_1, \dots, \mathcal{B}_{N+1}$ for one step each, and output the solution of

---

**Algorithm 4** Smoothed OGD (SOGD)

---

**Require:** Two parameters: $N$ and $Z$
1: Set $\mathcal{B}_0$ be any baseline
2: **for** $i = 1, \ldots, N+1$ **do**
3:    Let $\mathcal{A}_i$ be an instance of OGD with step size $\eta_i = \frac{D}{G}\sqrt{\frac{2^i}{T}}$
4:    Let $\mathcal{B}_i$ be an instance of Combiner, i.e., Algorithm 3 which combines $\mathcal{B}_{i-1}$ and $\mathcal{A}_i$ with parameters $\lambda_i = 1 - 2^{i-1}/T$ and $Z$
5: **end for**
6: **for** $t = 1, \ldots, T$ **do**
7:    Run $\mathcal{B}_1, \ldots, \mathcal{B}_{N+1}$ sequentially for one step each
8:    Output the solution of $\mathcal{B}_{N+1}$, denoted by $\mathbf{w}_t$
9: **end for**

---

$\mathcal{B}_{N+1}$. It is important to construct the sequence of algorithms $\mathcal{A}_i/\mathcal{B}_i$ in *descending* order of their associated discount factors, as this ordering is crucial for analyzing the overall discounted regret of the algorithm. Finally, we leverage a technical lemma (Kapralov & Panigrahy, 2010, Lemma 20) to extend the results from a discrete set to a continuous interval. Following Zhang et al. (2022a), we refer to this algorithm as Smoothed OGD (SOGD), and summarize it in Algorithm 4.

We have the following theoretical guarantee regarding the discounted regret of SOGD.

**Theorem 3.** *Suppose $\tau \geq \max\{16e, 32\log\frac{1}{Z}\}$ and set $Z = 1/T$. Under Assumptions 1, 2 and 3, for any $\lambda \in [1 - 1/\tau, 1 - 1/T]$, Algorithm 4 satisfies*

$$\sum_{t=1}^{T} \lambda^{T-t} f_t(\mathbf{w}_t) - \sum_{t=1}^{T} \lambda^{T-t} f_t(\mathbf{w}) \leq \frac{2GD}{\sqrt{1-\lambda}}\left(4\sqrt{\log T} + \sqrt{2}\right) + \frac{GD(N+1)}{(1-\lambda)T} + 2GD$$

*for any $\mathbf{w} \in \mathcal{W}$, where $N = \left\lceil \log_2 \frac{T}{\tau} \right\rceil$.*

**Remark:** Theorem 3 shows that SOGD achieves an $O(\sqrt{\log T/(1-\lambda)})$ bound for $\lambda$-discounted regret, holding simultaneously for all $\lambda \in [1 - 1/\tau, 1 - 1/T]$. Compared to the $O\left(1/\sqrt{1-\lambda}\right)$ guarantee in Theorem 1 for a known discount factor $\lambda$, this bound incurs an additional $O(\sqrt{\log T})$ factor, reflecting the cost of adaptivity to the discount factor. We emphasize that this additional term is fundamental, as existing algorithms for adaptive regret also incur such $\log T$ term (see Appendix B).

## 4   EXPERIMENT

In this section, we conduct empirical experiments to validate the effectiveness of our methods.

**Setup and contenders**   Our experiments are conducted on the ijcnn1 dataset from LIBSVM Data (Chang & Lin, 2011; Prokhorov, 2001), and the dimension of features is $d = 22$. We study an online classification problem where, at each round $t \in [T]$, the learner selects a model $\mathbf{w}_t \in \mathcal{W}$. After submitting this decision, the learner receives a mini-batch of training examples $\{(x_t^{(i)}, y_t^{(i)})\}_{i=1}^{m}$, where each pair consists of a feature vector and its label, drawn from a distribution. The learner evaluates performance using a convex loss function $f_t(\mathbf{w}_t)$, which reflects the loss incurred, and then updates the model accordingly. In this work, we set the total rounds $T = 10000$, and the online learner suffers the absolute loss. We evaluate the performance of our proposed method (SOGD) by comparing it with four baselines, including OGD using a tuned step size in Theorem 1, strongly adaptive algorithms (SAOL (Daniely et al., 2015) and SCB (Jun et al., 2017)), and a untuned variant of OGD. For a fair comparison, all methods are implemented under identical experimental settings.

**Results**   We repeat the experiments for five times and evaluate the algorithms using the discounted cumulative loss with different factors ranging from 0.5 to 0.999. We report a subset of the results here, with the remaining results provided in Appendix D. As illustrated in Figure 3, our proposed SOGD algorithm achieves performance comparable to that of OGD with a specifically tuned step size. Notably, for $\lambda = 0.999$ and 0.998, both algorithms exhibit similar loss accumulation. For $\lambda = 0.996$,

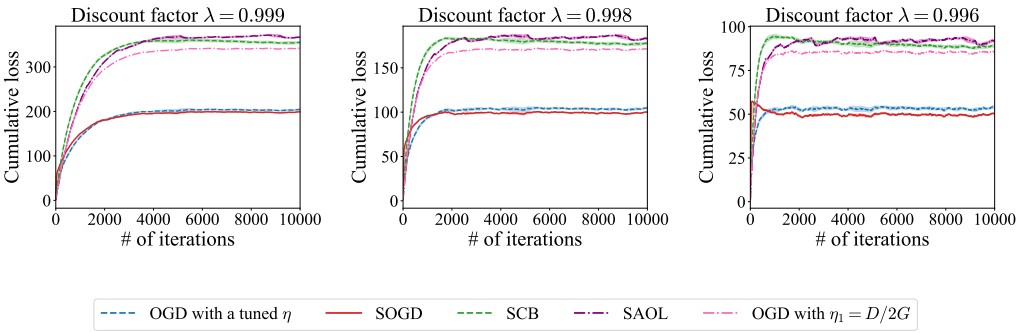

Figure 3: Performance comparison of discounted cumulative loss with different discount factors.

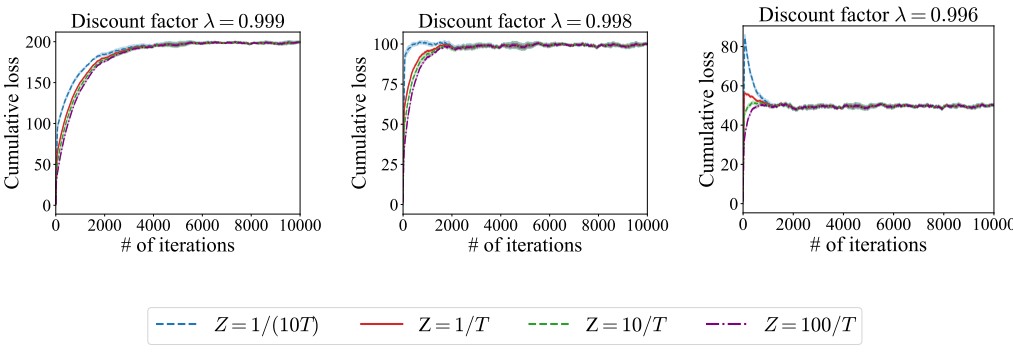

Figure 4: Parameter sensitivity on different choices of $Z$ with different discount factors.

while OGD shows slightly lower initial loss, SOGD quickly converges to a comparable steady-state loss. Furthermore, SOGD does not require prior knowledge of $\lambda$ for its configuration, offering the advantage of adaptability in scenarios where the discount factor is unknown.

**Parameter sensitivity**  We also conduct a parameter sensitivity study, as our proposed SOGD algorithm involves two tunable parameters, $Z$ and $\tau$. We report the results for different choices of $Z$ here, while the $\tau$ is provided in Appendix D. As shown in Figure 4, our empirical findings demonstrate that SOGD is not highly sensitive to the choice of $Z$ within a reasonable range that aligns with our theoretical results. All four variants of SOGD with different $Z$ achieve similarly strong performance, indicating the robustness of our method.

## 5  CONCLUSION AND FUTURE WORK

In this paper, we study discounted OCO with a discount factor $\lambda$. First, we investigate OGD with constant step size, and prove that OGD with step size $\eta = O(\sqrt{1-\lambda})$ achieves an $O(1/\sqrt{1-\lambda})$ bound for $\lambda$-discounted regret. Second, we focus on the challenging case with an unknown discount factor. To establish discounted regret for all possible factors across a continuous interval, we provide a novel analysis on the discounted payoff of DNP-cu, and show that DNP-cu enables aggregation of two experts with different discount factors. Thus, we employ DNP-cu to sequentially aggregate the decisions from experts with different configurations to achieve our goal.

There are several directions for future research. First, it is common to consider the case where the discount factor is time-varying. Extending our framework to the time-varying case is an important but challenging direction. Second, we can obtain tighter bounds for classical OCO when the functions are strongly convex or exp-concave. It remains open whether such curvature properties can be exploited to yield improved discounted regret.

ACKNOWLEDGMENTS

This work was partially supported by NSFC (U23A20382, 62361146852), and the Collaborative Innovation Center of Novel Software Technology and Industrialization.

REPRODUCIBILITY STATEMENT

The contributions of this paper are mainly theoretical. We clearly state the problem setting and all assumptions used in our analysis, and provide complete proofs of the theoretical results in the appendix. For the experimental part, we describe in detail the publicly available datasets used, as well as the experimental setup, including parameter configurations, baselines, and other implementation details. These components together ensure that both the theoretical and empirical findings of our work can be independently verified and reproduced.

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

# A ANALYSIS

In this section, we present the analysis of all theorems and lemmas.

## A.1 PROOF OF THEOREM 1

The proof is similar to that of Zhang et al. (2024, Theorem 6), except that we adopt a different step size. From Jensen's inequality and the standard analysis of OGD, we have

$$
\sum_{t=1}^{T} \lambda^{T-t} f_t(\mathbf{w}_t) - \sum_{t=1}^{T} \lambda^{T-t} f_t(\mathbf{w})
$$
$$
\leq \sum_{t=1}^{T} \lambda^{T-t} \langle \nabla f_t(\mathbf{w}_t), \mathbf{w}_t - \mathbf{w} \rangle
$$
$$
\leq \sum_{t=1}^{T} \frac{\lambda^{T-t}}{2\eta} \left( \|\mathbf{w}_t - \mathbf{w}\|^2 - \|\mathbf{w}_{t+1} - \mathbf{w}\|^2 \right) + \frac{\eta}{2} \sum_{t=1}^{T} \lambda^{T-t} \|\nabla f_t(\mathbf{w}_t)\|^2 \qquad (21)
$$
$$
\leq \frac{\lambda^{T-1}}{2\eta} \|\mathbf{w}_1 - \mathbf{w}\|^2 + \sum_{t=2}^{T} \left( \frac{\lambda^{T-t}}{2\eta} - \frac{\lambda^{T-(t-1)}}{2\eta} \right) \|\mathbf{w}_t - \mathbf{w}\|^2 + \frac{\eta}{2} \sum_{t=1}^{T} \lambda^{T-t} \|\nabla f_t(\mathbf{w}_t)\|^2
$$
$$
\leq \frac{\lambda^{T-1} D^2}{2\eta} + \frac{(1-\lambda) D^2}{2\eta} \sum_{t=2}^{T} \lambda^{T-t} + \frac{\eta G^2}{2} \sum_{t=1}^{T} \lambda^{T-t}.
$$

For any $\lambda \in (0,1)$, using $\sum_{t=1}^{T} \lambda^{T-t} \leq 1/(1-\lambda)$, (21) implies

$$
\sum_{t=1}^{T} \lambda^{T-t} f_t(\mathbf{w}_t) - \sum_{t=1}^{T} \lambda^{T-t} f_t(\mathbf{w}) \leq \frac{\lambda^{T-1} D^2}{2\eta} + \frac{(1-\lambda) D^2}{2\eta} \frac{1}{1-\lambda} + \frac{\eta G^2}{2} \frac{1}{1-\lambda} \leq \frac{D^2}{\eta} + \frac{\eta G^2}{2(1-\lambda)}.
$$

Setting

$$
\eta = \frac{D \sqrt{2(1-\lambda)}}{G}
$$

yields

$$
\sum_{t=1}^{T} \lambda^{T-t} f_t(\mathbf{w}_t) - \sum_{t=1}^{T} \lambda^{T-t} f_t(\mathbf{w}) \leq \frac{DG\sqrt{2}}{\sqrt{1-\lambda}}.
$$

Furthermore, when $\lambda \in [1 - 1/T, 1]$, we can alternatively set

$$
\eta = \frac{D}{G\sqrt{T}}.
$$

Since $\sum_{t=1}^{T} \lambda^{T-t} \leq T$, and

$$
\lambda^{T-1} + (1-\lambda) \sum_{t=2}^{T} \lambda^{T-t} \leq 2,
$$

(21) gives

$$
\sum_{t=1}^{T} \lambda^{T-t} f_t(\mathbf{w}_t) - \sum_{t=1}^{T} \lambda^{T-t} f_t(\mathbf{w}) \leq \frac{D^2}{\eta} + \frac{\eta G^2 T}{2} = \frac{3}{2} DG\sqrt{T}.
$$

This completes the proof.

## A.2 PROOF OF THEOREM 2

The proof builds on the approach of Zhang et al. (2022a, Theorem 1), and we highlight the main difference below.

Following the arguments of Kapralov & Panigrahy (2010) and Zhang et al. (2022a), we analyze the payoff of Algorithm 2 by leveraging Algorithm 1. This is because the update rule of Algorithm 1 is simpler, making the analysis more tractable. Specifically, we construct the following bit sequence

$$\tilde{b}_t = \begin{cases} b_t, & \text{if Line 6 of Algorithm 2 is executed at round } t; \\ 0, & \text{otherwise.} \end{cases}$$

It is straightforward to verify that the prediction $g(x_t)$, as well as the deviation $x_t$, generated by Algorithm 2 on the bit sequence $b_1, \ldots, b_T$, is exactly the same as that of Algorithm 1 on the transformed sequence $\tilde{b}_1, \ldots, \tilde{b}_T$. In the following, we first establish the theoretical guarantee for Algorithm 1 on the transformed sequence and then translate this result to the payoff of Algorithm 2 on the original sequence. We establish the following theorem for the discounted payoff of Algorithm 1.

**Theorem 4.** *Suppose $Z \leq \frac{1}{e}$ and $n \geq \max\{8e, 16\log\frac{1}{Z}\}$. For any bit sequence $b_1, \ldots, b_T$ such that $|b_t| \leq 1$, and any $\eta \geq \rho = 1 - 1/n$, the discounted payoff of Algorithm 1 satisfies*

$$\sum_{t=1}^{T} \eta^{T-t} g(x_t) b_t \geq \sum_{t=1}^{T} \frac{\eta^{T-t}}{n} \left( \frac{x_t g(x_t)}{2} - \Phi_t \right) - \frac{Z}{2(1-\eta)} \tag{22}$$

*where $\Phi_t = \int_0^{x_t} g(s)ds$ is the potential function. Furthermore,*

$$\sum_{t=1}^{T} \rho^{T-t} g(x_t) b_t \geq \sum_{t=1}^{T} \rho^{T-t} b_t + \sum_{t=1}^{T} \frac{\rho^{T-t}}{n} \left( \frac{x_t g(x_t)}{2} - \Phi_t \right) - \frac{Z}{2(1-\rho)} - x_{T+1}. \tag{23}$$

From Theorem 4, we have

$$\sum_{t=1}^{T} \eta^{T-t} g(x_t) \tilde{b}_t \geq \sum_{t=1}^{T} \frac{\eta^{T-t}}{n} \left( \frac{x_t g(x_t)}{2} - \Phi_t \right) - \frac{Z}{2(1-\eta)}, \tag{24}$$

$$\sum_{t=1}^{T} \rho^{T-t} g(x_t) \tilde{b}_t \geq \sum_{t=1}^{T} \rho^{T-t} \tilde{b}_t + \sum_{t=1}^{T} \frac{\rho^{T-t}}{n} \left( \frac{x_t g(x_t)}{2} - \Phi_t \right) - \frac{Z}{2(1-\rho)} - x_{T+1}. \tag{25}$$

From (38) of Zhang et al. (2022a), we have

$$g(x_k)(b_k - \tilde{b}_k) \geq 0, \text{ if } b_k \neq \tilde{b}_k, \tag{26}$$

$$g(x_k)(b_k - \tilde{b}_k) \geq b_k - \tilde{b}_k, \text{ if } b_k \neq \tilde{b}_k. \tag{27}$$

which implies

$$\begin{aligned} \sum_{t=1}^{T} \eta^{T-t} g(x_t) b_t &= \sum_{t=1}^{T} \eta^{T-t} g(x_t) \tilde{b}_t + \sum_{t=1}^{T} \eta^{T-t} g(x_t)(b_t - \tilde{b}_t) \\ &\overset{(24),(26)}{\geq} \sum_{t=1}^{T} \frac{\eta^{T-t}}{n} \left( \frac{x_t g(x_t)}{2} - \Phi_t \right) - \frac{Z}{2(1-\eta)}, \end{aligned} \tag{28}$$

$$\begin{aligned} \sum_{t=1}^{T} \rho^{T-t} g(x_t) b_t &= \sum_{t=1}^{T} \rho^{T-t} g(x_t) \tilde{b}_t + \sum_{t=1}^{T} \rho^{T-t} g(x_t)(b_t - \tilde{b}_t) \\ &\overset{(25),(27)}{\geq} \sum_{t=1}^{T} \rho^{T-t} b_t + \sum_{t=1}^{T} \frac{\rho^{T-t}}{n} \left( \frac{x_t g(x_t)}{2} - \Phi_t \right) - \frac{Z}{2(1-\rho)} - x_{T+1}. \end{aligned} \tag{29}$$

To simplify the term involving $\frac{x_t g(x_t)}{2} - \Phi_t$, we make use of the following property of Algorithm 1 (Zhang et al., 2022a, (33))

$$-1 \leq x_t \leq U(n) + 1, \ \forall t \geq 1 \tag{30}$$

and establish the following lemma.

**Lemma 1.** *Suppose $Z \leq \frac{1}{e}$, $n \geq \max\{8e, 16\log\frac{1}{Z}\}$ and $U(n) \geq 20$. Under the condition in (30), we have*

$$\Phi_t = \int_0^{x_t} g(s)ds \leq \frac{x_t g(x_t)}{2}. \tag{31}$$

Finally, we obtain (13) by combining (28) and (31), and obtain (14) by combining (29), (30) and (31).

### A.3 PROOF OF THEOREM 3

We start by presenting the discounted regret of each OGD algorithm, namely $\mathcal{A}_i$. Let $\mathbf{w}_{t,i}$ be the output of $\mathcal{A}_i$ at round $t$, and recall that the step size is set according to (12). From Theorem 1, we have

$$\sum_{t=1}^{T} \lambda_i^{T-t} f_t(\mathbf{w}_{t,i}) - \sum_{t=1}^{T} \lambda_i^{T-t} f_t(\mathbf{w}) \le \frac{DG\sqrt{2}}{\sqrt{1-\lambda_i}} \tag{32}$$

for all $\mathbf{w} \in \mathcal{W}$, where $\lambda_i = 1 - 2^{i-1}/T$.

Next, we proceed to bound the discounted regret of SOGD at each discount factor $\lambda_i$, where $i \in [N+1]$. To this end, we make use of the theoretical guarantee of DNP-cu stated in Theorem 2. Note that the conditions of Theorem 2 impose a lower bound on the window size $n$. Therefore, to apply this result, we need to ensure that the minimal window size appearing in Algorithm 4 satisfies

$$\min_{i \in [N+1]} \frac{1}{1-\lambda_i} = \frac{T}{2^N} \ge \max\left\{8e, 16\log\frac{1}{Z}\right\}. \tag{33}$$

Since

$$\frac{T}{2^N} \ge \frac{T}{2^{1+\log_2\frac{T}{\tau}}} = \frac{\tau}{2},$$

(33) holds when $\tau \ge \max\{16e, 32\log\frac{1}{Z}\}$. Furthermore, we also require $U(n) \ge 22$ in Theorem 2. We now proceed to estimate a lower bound for $U(n)$. Since $g(x)$ is a monotonically increasing function on $[0, U(n)]$, it follows that if we can find a point $x'$ such that $g(x') < 1$, then we must have $x' < U(n)$. Setting $x = 4\sqrt{n}$, we have

$$g(4\sqrt{n}) = \sqrt{\frac{n}{8}} Z \cdot \text{erf}\left(\frac{1}{\sqrt{2}}\right) e \le \frac{\sqrt{\pi}}{4\sqrt{2}} e \approx 0.8517$$

where the last step is because we set $Z = 1/T$ and $n = \tau \le T$, and property of the error function. Since $g(4\sqrt{n}) < 1$, we have

$$U(n) \ge 4\sqrt{n} \ge 4\sqrt{32} \ge 22.$$

Denote the output of $\mathcal{B}_i$ at round $t$ by $\mathbf{v}_{t,i}$, and observe that the final output is $\mathbf{w}_t = \mathbf{v}_{t,N+1}$. Then, we decompose the $\lambda_i$-discounted regret of SOGD as

$$\sum_{t=1}^{T} \lambda_i^{T-t} f_t(\mathbf{w}_t) - \sum_{t=1}^{T} \lambda_i^{T-t} f_t(\mathbf{w}) = \sum_{t=1}^{T} \lambda_i^{T-t} f_t(\mathbf{v}_{t,N+1}) - \sum_{t=1}^{T} \lambda_i^{T-t} f_t(\mathbf{w})$$

$$= \sum_{k=i+1}^{N+1} \left( \underbrace{\sum_{t=1}^{T} \lambda_i^{T-t} f_t(\mathbf{v}_{t,k}) - \lambda_i^{T-t} f_t(\mathbf{v}_{t,k-1})}_{\alpha_k} \right) + \underbrace{\sum_{t=1}^{T} \lambda_i^{T-t} f_t(\mathbf{v}_{t,i}) - \sum_{t=1}^{T} \lambda_i^{T-t} f_t(\mathbf{w}_{t,i})}_{\beta} \tag{34}$$

$$+ \underbrace{\sum_{t=1}^{T} \lambda_i^{T-t} f_t(\mathbf{w}_{t,i}) - \sum_{t=1}^{T} \lambda_i^{T-t} f_t(\mathbf{w})}_{\gamma}$$

where $\alpha_k$ denotes the $\lambda_i$-discounted regret of $\mathcal{B}_k$ with respect to $\mathcal{B}_{k-1}$, $\beta$ denotes the $\lambda_i$-discounted regret of $\mathcal{B}_i$ with respect to $\mathcal{A}_i$, and $\gamma$ denotes the $\lambda_i$-discounted regret of $\mathcal{A}_i$. We can directly bound $\gamma$ using (32), so we next focus on bounding $\alpha_k$ for $k = i+1, \ldots, N+1$, and $\beta$.

Recall that $\mathcal{B}_k$ invokes DNP-cu with discount factor $\lambda_k$ to aggregate $\mathcal{B}_{k-1}$ and $\mathcal{A}_k$. Define

$$\ell_{t,k} = \frac{f_t(\mathbf{v}_{t,k-1}) - f_t(\mathbf{w}_{t,k})}{GD}.$$

Following the derivation of (18), we have

$$\alpha_k = \sum_{t=1}^{T} \lambda_i^{T-t} f_t(\mathbf{v}_{t,k}) - \lambda_i^{T-t} f_t(\mathbf{v}_{t,k-1}) \le -GD \sum_{t=1}^{T} \lambda_i^{T-t} \omega_{t,k} \ell_{t,k} \tag{35}$$

where $\omega_{t,k}$ is the output of the $k$-th instance of DNP-cu at round $t$. Since the discount factors are set in descending order in Algorithm 4, it holds that

$$\lambda_i > \lambda_k, \text{ for all } k = i+1, \ldots, N+1.$$

Thus, we can use (13) in Theorem 2 to upper bound the RHS of (35), yielding

$$\alpha_k \overset{(13),(35)}{\leq} \frac{GDZ}{2(1-\lambda_i)}. \tag{36}$$

Similarly, following the derivation of (19), we have

$$\beta = \sum_{t=1}^{T} \lambda_i^{T-t} f_t(\mathbf{v}_{t,i}) - \sum_{t=1}^{T} \lambda_i^{T-t} f_t(\mathbf{w}_{t,i}) \leq -GD \sum_{t=1}^{T} \lambda_i^{T-t} \left( \omega_{t,i} \ell_{t,i} - \ell_{t,i} \right)$$

$$\overset{(14)}{\leq} GD \left( \frac{Z}{2(1-\lambda_i)} + U\left(\frac{1}{1-\lambda_i}\right) + 1 \right) \overset{(15)}{\leq} GD \left( \frac{Z}{2(1-\lambda_i)} + \sqrt{16 \frac{1}{1-\lambda_i} \log \frac{1}{Z}} + 1 \right). \tag{37}$$

Substituting (32), (36) and (37) into (34), we obtain

$$\sum_{t=1}^{T} \lambda_i^{T-t} f_t(\mathbf{w}_t) - \sum_{t=1}^{T} \lambda_i^{T-t} f_t(\mathbf{w}) \leq \frac{GD}{\sqrt{1-\lambda_i}} \left( 4\sqrt{\log \frac{1}{Z}} + \sqrt{2} \right) + \frac{GD(N+1)Z}{2(1-\lambda_i)} + GD \tag{38}$$

for all $i = 1, \ldots, N+1$.

So far, we have established discounted regret guarantees for each discount factor $\lambda_i$, where $i \in [N+1]$. Next, we extend these results to the continuous range of $\lambda$ values specified in (10). To this end, we make use of the following lemma (Kapralov & Panigrahy, 2010, Lemma 20).

**Lemma 2.** *Given a sequence $s_1, \ldots, s_T$, and a discount factor $\lambda < 1$, define the $\lambda$-smoothed average at time $T$ by*

$$s_T^\lambda = (1-\lambda) \sum_{t=1}^{T} \lambda^{T-t} s_t.$$

*Then, for any $\lambda_1 > \lambda_2$, the $\lambda_1$-smoothed average at time $T$ is a convex combination of $\lambda_2$-smoothed average at time $j \leq T$:*

$$s_T^{\lambda_1} = \frac{1-\lambda_1}{1-\lambda_2} s_T^{\lambda_2} + \sum_{j<T} \frac{(1-\lambda_1)(\lambda_1-\lambda_2)\lambda_1^{T-j-1}}{1-\lambda_2} s_{T-j}^{\lambda_2}$$

*where the coefficients on the RHS are nonnegative and sum to $1$.*

According to our construction in (11), for each $\lambda \in [1 - 1/\tau, 1 - 1/T]$, there exists an index $i \in [N]$ such that

$$\lambda_{i+1} = 1 - \frac{2^i}{T} \leq \lambda \leq \lambda_i = 1 - \frac{2^{i-1}}{T}$$

implying

$$\frac{1}{1-\lambda_{i+1}} \leq \frac{1}{1-\lambda}, \text{ and } \frac{1-\lambda_{i+1}}{1-\lambda} \leq 2. \tag{39}$$

From Lemma 2, we know that

$$(1-\lambda) \sum_{t=1}^{T} \lambda^{T-t} \left[ f_t(\mathbf{w}_t) - f_t(\mathbf{w}) \right]$$

can be expressed as a convex combination of

$$(1-\lambda_{i+1}) \sum_{t=1}^{T} \lambda_{i+1}^{T-t} \left[ f_t(\mathbf{w}_t) - f_t(\mathbf{w}) \right], (1-\lambda_{i+1}) \sum_{t=1}^{T-1} \lambda_{i+1}^{T-1-t} \left[ f_t(\mathbf{w}_t) - f_t(\mathbf{w}) \right], \ldots.$$

As a result

$$(1 - \lambda) \sum_{t=1}^{T} \lambda^{T-t} \big[ f_t(\mathbf{w}_t) - f_t(\mathbf{w}) \big]$$

$$\leq (1 - \lambda_{i+1}) \max_{j \in [T]} \sum_{t=1}^{j} \lambda_{i+1}^{j-t} \big[ f_t(\mathbf{w}_t) - f_t(\mathbf{w}) \big]$$

$$\overset{(38)}{\leq} (1 - \lambda_{i+1}) \left[ \frac{GD}{\sqrt{1 - \lambda_{i+1}}} \left( 4\sqrt{\log \frac{1}{Z}} + \sqrt{2} \right) + \frac{GD(N+1)Z}{2(1 - \lambda_{i+1})} + GD \right]$$

where we use the fact that (38) holds for any integer $T$. Thus, we have

$$\sum_{t=1}^{T} \lambda^{T-t} \big[ f_t(\mathbf{w}_t) - f_t(\mathbf{w}) \big]$$

$$\leq \frac{1 - \lambda_{i+1}}{1 - \lambda} \left[ \frac{GD}{\sqrt{1 - \lambda_{i+1}}} \left( 4\sqrt{\log \frac{1}{Z}} + \sqrt{2} \right) + \frac{GD(N+1)Z}{2(1 - \lambda_{i+1})} + GD \right]$$

$$\overset{(39)}{\leq} \frac{2GD}{\sqrt{1 - \lambda}} \left( 4\sqrt{\log \frac{1}{Z}} + \sqrt{2} \right) + \frac{GD(N+1)Z}{1 - \lambda} + 2GD.$$

Finally, we set $Z = 1/T$ to finish the proof.

### A.4 PROOF OF THEOREM 4

The overall proof strategy is similar to that of Zhang et al. (2022a, Theorem 5), with the key distinction being that we consider the discounted payoff rather than the standard payoff. Moreover, our proof is more concise, as we focus on the interval $[1, T]$ and omit the switching cost. Building upon their analysis while adapting it to our setting, we directly leverage several intermediate results to avoid redundancy.

From (70) of Zhang et al. (2022a), we know the difference between any two consecutive deviations is bounded by 2, i.e.,

$$|x_t - x_{t+1}| \leq 2. \tag{40}$$

Let $\mathbb{I}(x)$ be the indicator function of the interval $[0, U(n) + 2]$. We have (Zhang et al., 2022a, (74))

$$4 \max_{s \in [x_t, x_{t+1}]} |g'(s)| \leq \frac{1}{n} x_t g(x_t) \mathbb{I}(x_t) + Z. \tag{41}$$

We also recall the following inequality for piecewise differentiable functions $f : [a, b] \mapsto \mathbb{R}$ (Kapralov & Panigrahy, 2010; Daniely & Mansour, 2019):

$$\int_a^b f(x) dx \leq f(a)(b - a) + \max |f'(z)| \frac{1}{2}(b - a)^2. \tag{42}$$

Following the idea outlined in Kapralov & Panigrahy (2010, proof of Lemma 18), we have

$$\Phi_{t+1} - \eta \Phi_t = \Phi_{t+1} - \Phi_t + (1 - \eta)\Phi_t = \int_{x_t}^{x_{t+1}} g(s) ds + (1 - \eta)\Phi_t$$

$$\overset{(42)}{\leq} g(x_t)(x_{t+1} - x_t) + \frac{1}{2}(x_{t+1} - x_t)^2 \max_{s \in [x_t, x_{t+1}]} |g'(s)| + (1 - \eta)\Phi_t$$

$$\overset{(40)}{\leq} g(x_t) \left( -\frac{1}{n} x_t + b_t \right) + 2 \max_{s \in [x_t, x_{t+1}]} |g'(s)| + (1 - \eta)\Phi_t \tag{43}$$

$$\overset{(41)}{\leq} g(x_t) \left( -\frac{1}{n} x_t + b_t \right) + \frac{1}{2n} x_t g(x_t) \mathbb{I}(x_t) + \frac{Z}{2} + (1 - \eta)\Phi_t$$

$$= g(x_t) b_t + \frac{1}{2n} x_t g(x_t) \left( \mathbb{I}(x_t) - 1 \right) + (1 - \eta)\Phi_t - \frac{1}{2n} x_t g(x_t) + \frac{Z}{2}$$

$$\leq g(x_t) b_t + \frac{1}{2n} x_t g(x_t) \left( \mathbb{I}(x_t) - 1 \right) + \frac{1}{n} \left( \Phi_t - \frac{x_t g(x_t)}{2} \right) + \frac{Z}{2}$$

where the last line follows from the condition $\eta \geq 1 - 1/n$. Then, we have

$$0 \leq \Phi_{T+1} = \Phi_{T+1} - \eta^T \Phi_1 = \sum_{t=1}^{T} \eta^{T-t} \left( \Phi_{t+1} - \eta \Phi_t \right)$$

$$\overset{(43)}{\leq} \sum_{t=1}^{T} \eta^{T-t} g(x_t) b_t + \sum_{t=1}^{T} \frac{1}{2n} \eta^{T-t} x_t g(x_t) \left( \mathbb{I}(x_t) - 1 \right) + \sum_{t=1}^{T} \frac{\eta^{T-t}}{n} \left( \Phi_t - \frac{x_t g(x_t)}{2} \right) + \frac{Z}{2} \sum_{t=1}^{T} \eta^{T-t}$$

$$\leq \sum_{t=1}^{T} \eta^{T-t} g(x_t) b_t + \sum_{t=1}^{T} \frac{1}{2n} \eta^{T-t} x_t g(x_t) \left( \mathbb{I}(x_t) - 1 \right) + \sum_{t=1}^{T} \frac{\eta^{T-t}}{n} \left( \Phi_t - \frac{x_t g(x_t)}{2} \right) + \frac{Z}{2(1-\eta)}$$
$$(44)$$

where in the first equality we use the fact that $\Phi_1 = 0$ since $x_1 = 0$.

We proceed to bound the second term in the last line of (44). Combining the simple observation that

$$x_t g(x_t) \left( 1 - \mathbb{I}(x_t) \right) \geq 0,$$

with (44), we obtain (22). Recall that

$$x_{T+1} = \sum_{t=1}^{T} \rho^{T-t} b_t$$

and thus

$$\sum_{t=1}^{T} \rho^{T-t} g(x_t) b_t \overset{(22)}{\geq} \sum_{t=1}^{T} \rho^{T-t} b_t + \sum_{t=1}^{T} \frac{\rho^{T-t}}{n} \left( \frac{x_t g(x_t)}{2} - \Phi_t \right) - \frac{Z}{2(1-\rho)} - x_{T+1}.$$

### A.5 PROOF OF LEMMA 1

We note that Kapralov & Panigrahy (2010, Lemma 22) have established a similar result for their confidence function in (3), but provided only an incomplete proof. In this paper, we adopt a novel and insightful approach to provide an entirely new and complete analysis.

First, when $x_t \in [-1, 0]$, we have

$$\Phi_t = 0 \leq \frac{x_t g(x_t)}{2} = 0.$$

since $g(s) = 0$ for $s \in [-1, 0]$. Second, when $x_t \in [0, U(n)]$, it follows from Daniely & Mansour (2019, Lemma 18) that $g(s)$ is convex over $[0, x_t]$. Therefore, (31) holds by the convexity of $g(s)$. Third, when $x_t \in [U(n), U(n) + 1]$, we have

$$\Phi_t = \int_0^{x_t} g(s) ds = \int_0^{U(n)} g(s) ds + x_t - U(n)$$

According to definition of $g(x)$, we have $\frac{x_t g(x_t)}{2} = \frac{x_t}{2}$ when $x_t \in [U(n), U(n) + 1]$. Accordingly, our objective reduces to proving the following inequality:

$$\Phi_t = \int_0^{U(n)} g(s) ds + x_t - U(n) \leq \frac{x_t}{2}$$

for all $x_t \in [U(n), U(n) + 1]$, which implies

$$\int_0^{U(n)} g(s) ds \leq U(n) - \frac{x_t}{2}. \tag{45}$$

To establish that (45) holds for all $x_t \in [U(n), U(n) + 1]$, it suffices to prove the inequality for the maximum value of $x_t$ in this interval, that is, when $x_t = U(n) + 1$. Specifically, we need to show that

$$\int_0^{U(n)} g(s) ds \leq \frac{U(n) - 1}{2}. \tag{46}$$

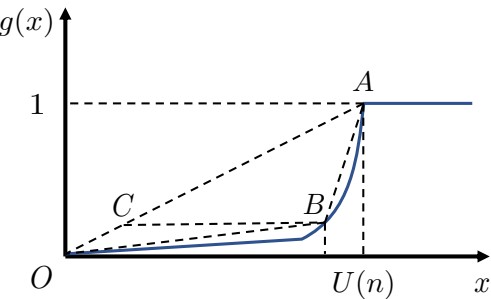

Figure 5: The confidence function $g(x)$.

As long as (46) can be established, the proof of Lemma 1 is complete. However, proving (46) remains highly challenging. To address this, we propose a novel approach that combines analytical and geometric insights. The confidence function $g(x)$ is illustrated in Figure 5.

Notably, we identify a point $B : (c, g(c))$ on the function $g(x)$. If we can show that the following inequality holds at this point, then (46) is immediately satisfied:

$$\int_0^{U(n)} g(s)ds + S_{\triangle OAB} \leq \frac{U(n)}{2}, \text{ and } S_{\triangle OAB} \geq \frac{1}{2}.$$

From Figure 5, it can be seen that the area of triangle $\triangle OAB$ is equal to the sum of the areas of triangles $\triangle OBC$ and $\triangle ABC$. Therefore, we have

$$S_{\triangle OAB} = \frac{1}{2}BC = \frac{1}{2}\left(c - U(n) \cdot g(c)\right)$$

where the point $C$ is $(U(n) \cdot g(c), g(c))$ because this point is on the line $OA : y = \frac{x}{U(n)}$. To prove (46), we need to find a point $B : (c, g(c))$ that it satisfies

$$c - U(n) \cdot g(c) \geq 1.$$

To achieve this, we find a specific point $B$ such that $c = U(n) - 8$, and we have

$$
\begin{aligned}
g(U(n) - 8) &= \sqrt{\frac{n}{8}} Z \cdot \mathrm{erf}\left(\frac{U(n) - 8}{\sqrt{8n}}\right) e^{\frac{(U(n)-8)^2}{16n}} \\
&\leq \sqrt{\frac{n}{8}} Z \cdot \mathrm{erf}\left(\frac{U(n)}{\sqrt{8n}}\right) e^{\frac{(U(n)-8)^2}{16n}} \\
&= e^{\frac{64 - 16U(n)}{16n}} \\
&\leq e^{\frac{64 - 16U(n)}{512}}
\end{aligned}
$$

where the last two steps are due to the definition of $U(n)$ and $n \geq 32$. Then, we find that

$$
\begin{aligned}
c - U(n) \cdot g(c) &= U(n) - 8 - U(n) \cdot g(U(n) - 8) \\
&\geq U(n)\left(1 - e^{\frac{64 - 16U(n)}{512}}\right) - 8 \\
&\geq 1.4648
\end{aligned}
$$

where the last step is due to $U(n) \geq 22$. We finish the proof.

## B FURTHER CLARIFICATIONS ON OPTIMALITY

Compared to the theoretical guarantee of OGD in Theorem 1, SOGD incurs an additional $\log T$ factor. We emphasize that this additional term is a necessary cost for adapting to an unknown parameter. As discussed in our work, discounted OCO with an unknown discount factor serves a

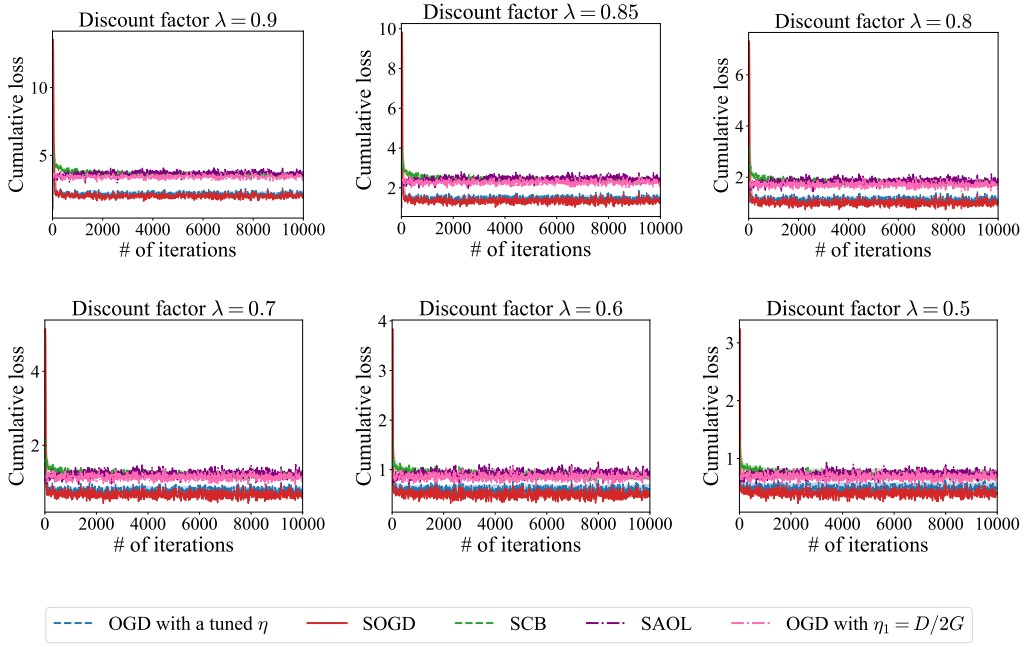

Figure 6: Performance comparison of discounted cumulative loss with different discount factors.

similar purpose to adaptive regret, as both effectively define a temporal horizon of interest. In fact, existing algorithms for adaptive regret (Hazan & Seshadhri, 2007; Jun et al., 2017; Zhang et al., 2018b) also incur such a term, as in our work. Specifically, several online algorithms achieve $O(\sqrt{\tau \log T})$, $O(\frac{1}{\lambda} \log \tau \log T)$, and $O(\frac{d}{\alpha} \log \tau \log T)$ for general convex, $\lambda$-strongly convex, and $\alpha$-exp-concave functions, respectively, where $\tau$ is the interval length. Compared to the minimax optimal result for static regret (Ordentlich & Cover, 1998; Abernethy et al., 2008), i.e., $O(\sqrt{T})$, $O(\frac{1}{\lambda} \log T)$, and $O(\frac{d}{\alpha} \log T)$, it is evident that the adaptive regret bounds suffer an additional $O(\sqrt{\log T})$ or $O(\log T)$ term, reflecting the cost of adaptivity to every interval.

## C  MORE DISCUSSION ON TECHNICAL CONTRIBUTIONS

In the literature, DNP-cu is not a well-studied algorithm. Kapralov & Panigrahy (2010) proposed the original algorithm, but without a rigorous theoretical analysis. Subsequent works by Daniely & Mansour (2019) and Zhang et al. (2022a) have progressively improved its theoretical foundations. To adapt it to discounted OCO with an unknown factor, we need to make technical contributions to the original theoretical analysis.

Specifically, we provide a new theoretical guarantee (Theorem 2) to demonstrate that DNP-cu can support $\lambda$-discounted regret when the parameter $\rho = \lambda$ is set, and at the same time, it can provide a guarantee on the discounted payoff for a different discount factor, which proves the theoretical feasibility of the algorithm design. Compared to the standard payoff analysis, we show that the discounted payoff with a different factor $\eta \geq \rho$ can be effectively bounded. Furthermore, we need to struggle with the potential function in Theorem 4. Bounding this potential function is particularly challenging, and previous work does not provide a rigorous proof. In our work, we establish a new lemma, i.e., Lemma 1, by adopting a novel and insightful approach to provide an entirely new and complete analysis to upper bound this function.

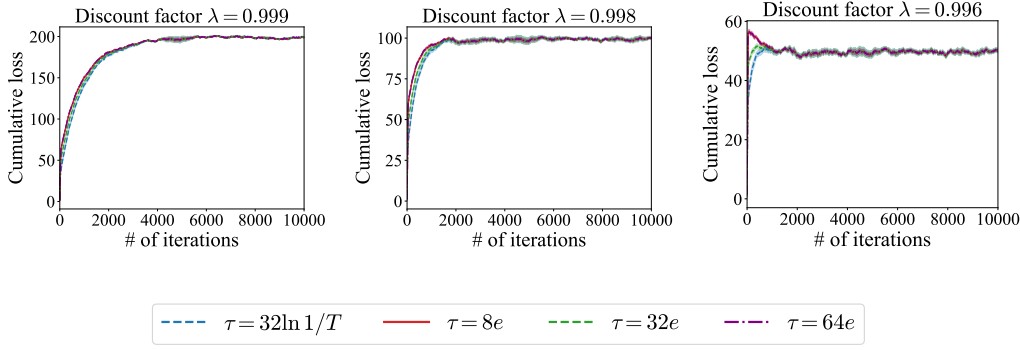

Figure 7: Parameter sensitivity on different choices of $\tau$ with different discount factors.

## D  ADDITIONAL EXPERIMENTS

Following the experimental setup in the main text, we additionally report experimental results across a wide range of discount factors $\lambda \in \{0.9, 0.85, 0.8, 0.7, 0.6, 0.5\}$. From the definition of discounted cumulative loss $\sum_{t=1}^{T} \lambda^{T-t} f_t(\mathbf{w}_t)$, it is clear that as the discount factor $\lambda$ decreases, our algorithm places more emphasis on the recent losses. As shown in Figure 6, all algorithms quickly converge to a stable minimum loss. For all values of discount factors $\lambda$, it is evident that our algorithm, SOGD, along with OGD with a tuned step size, outperforms other baselines.

We also conduct experiments on different choices of $\tau$. As shown in Figure 7, the performance of SOGD does not vary significantly and converges steadily to the optimal solution. Combined with the parameter sensitivity experiments for choices of $Z$ in the main text, we recommend setting $Z = 1/T$ and $\tau = 32 \ln(1/T)$ or $32e$ for practical guidelines.

## E  PRACTICAL MOTIVATIONS FOR THE UNKNOWN DISCOUNT FACTOR

In many real-world applications, the discount factor $\lambda$ cannot be directly specified by the designer, yet selecting an appropriate "true" value is crucial for reliable decision-making. We highlight two representative scenarios where the discount factor inherently exists but is unknown in practice.

- **Control Systems:** In many control systems, including LQR and other infinite-horizon formulations, the discount factor implicitly reflects physical properties of the system, such as responsiveness, energy dissipation, degradation, and long-term stability. In realistic scenarios, this discount factor is typically determined based on system dynamics, time delays, or reliability decay. Therefore, this factor cannot be directly chosen by the designer, as doing so would lead to unstable controlling.

- **Economic Decision Modeling:** In financial forecasting and portfolio optimization, the discount factor determines how rapidly past returns are devalued when assessing current market conditions. Its value is implicitly determined by latent market characteristics such as long-term dependence, autocorrelation strength, and structural stability. Thus, the discount factor is not an arbitrary design choice but an unknown parameter rooted in real market behavior, requiring careful estimation through empirical studies or behavioral assumptions.

## F  THE USE OF LLMS

In preparing this manuscript, we used large language models (LLMs) solely to improve the clarity and readability of the writing, with the goal of helping readers better understand our ideas and methods. LLMs did not contribute to any part of the research itself, including the generation of ideas, theoretical proofs, algorithm design, or other components that constitute the core contributions of this paper.

