# OpenReview forum: "Discounted Online Convex Optimization: Uniform Regret Across a Continuous Interval"
_ICLR.cc/2026/Conference — ICLR 2026 Poster_

### Official Review · Reviewer_uNa2 · 2025-10-30

**Soundness:** 2
**Presentation:** 3
**Contribution:** 3
**Rating:** 6
**Confidence:** 3

**Summary:**

This paper studies Discounted Online Convex Optimization (OCO) to minimize the λ-discounted regret. The
authors introduce a Smoothed OGD (SOGD) algorithm to achieve uniform discounted regret $O(\sqrt{\log(T)/(1-\lambda)})$
across a continuous interval of discount factors without prior knowledge of $\lambda$. They also provide a novel analysis of DNP-cu under discounted payoff settings, showing that it can combine experts operating under different discount factors.

**Strengths:**

1. The application of DNP-cu to combine experts with different discount factors is novel and the analysis of DNP-cu under discounted payoffs (Theorem 2) is a key contribution.

2. The paper provides a uniform regret bound across a continuous interval of $\lambda$ with detailed and well-organized proofs in the appendix.

3. The paper is well-structured with clear motivation, problem setup and technical exposition. The presentation of figures and algorithms help illustrate the main ideas.

4. The authors provide clear statement of problem setting and experimental setup, supporting the reproducibility of both the theoretical and empirical findings.

**Weaknesses:**

1. The experiments conducted in the paper only compare the performance of SOGD with typical OGD. It would be valuable to compare with other adaptive or meta-learning baselines.

2. Computational Practicality: The theoretical bounds of this paper depend on several constants and assumptions, which may lead to limited practicality and expensive computational cost in high-dimensional or large-scale settings.

**Questions:**

1. Can the problem be extended to settings with strongly convex or exp-concave losses? What regret bounds could be achieved?

2. How sensitive is SOGD to the choice of $Z$ and $\tau$? Are there practical guidelines for setting these parameters in real applications?

3. How does SOGD compare empirically and theoretically with state-of-the-art algorithms with strong adaptive regret when evaluated under discounted regret?

---

> ### Author Response · Authors · 2025-11-20
>
> **Many thanks for the constructive reviews! We have revised our paper accordingly, including more experiments on comparison with other baselines and parameter sensitivity, and future work on exp-concave or strongly-convex case. Please refer to our revised version for details.**
>
> ---
>
> **Q1:** More experiments on comparison with other adaptive baselines.
>
> **A1:** Thank you for the suggestion! We have added two representative strongly adaptive algorithms, SAOL and SCB, as baselines in the experimental section. Please refer to **Section 4 Experiment in our revised version** for details. As shown in Figure 3, our SOGD achieves better performance than the strongly adaptive algorithms. The strongly adaptive methods are not well suited for the discounted setting and therefore exhibit performance comparable to the OGD variant with untuned step size.
>
> ---
>
> **Q2:** Computational Practicality: The theoretical bounds of this paper depend on several constants and assumptions,which may lead to limited practicality and expensive computational cost in high-dimensional or large-scale settings.
>
> **A2:** We would like to clarify that the assumptions used in this work are standard in the OCO literature and are commonly employed in similar studies. Furthermore, we focus on an online setting, which ensures that the algorithm remains efficient and does not require excessive computational resources.
>
> ---
>
> **Q3:** Can the problem be extended to settings with strongly convex or exp-concave losses? What regret bounds could be achieved?
>
> **A3:** This is an insightful question. Unfortunately, the existing techniques are not sufficient to address the strongly convex or exp-concave cases in the discounted setting. Even in the simple scenario where the discount factor is known, it remains unclear how to exploit strong convexity or exp-concavity to obtain improved discounted regret guarantees. This suggests that extending discounted OCO beyond the convex setting is highly non-trivial.
>
> We regard this as a significant avenue for future exploration in discounted OCO, and have reflected this in the future work. Please refer to **Section 5 Future Work in our revised version** for details.
>
> ---
>
> **Q4:** How sensitive is SOGD to the choice of $Z$ and $\tau$? Are there practical guidelines for setting these parameters inreal applications?
>
> **A4:** Thank you for the suggestion. We have added experiments on parameter sensitivity with respect to the choice of $Z$ and $\tau$. Additional experiments show that as long as the parameters are chosen within a reasonable range (aligned with our theoretical results), the performance of SOGD does not vary significantly and converges steadily to the optimal solution. For practical guidelines, we recommend setting $Z=1/T$ and $\tau=32\ln (1/T)$ or $32e$. Please refer to **Section 4 Experiment in our revised version** for details.
>
> ---
>
> **Q5:** How does SOGD compare empirically and theoretically with state-of-the-art algorithms with strong adaptive regret when evaluated under discounted regret?
>
> **A5:** As addressed in Q1, we conduct empirical comparisons between SOGD and strongly adaptive algorithms. From a theoretical perspective, although adaptive regret and discounted regret are conceptually related, neither can be derived from the other. Consequently, strongly adaptive methods are unable to deliver regret guarantees in the discounted setting.

---

### Official Review · Reviewer_hZvg · 2025-10-31

**Soundness:** 3
**Presentation:** 3
**Contribution:** 3
**Rating:** 8
**Confidence:** 4

**Summary:**

The authors consider the setting of online convex optimization with $ \lambda $-discounted regret. Prior work assumes known $ \lambda $ and shows online gradient descent achieves $O ( (1-\lambda)^{-\frac12} ) $ discounted regret. In this work the authors consider the setting of \emph{unknown} $ \lambda $ and show that a carefully designed aggregation of online gradient descent instances can achieve $ O ( \log T (1-\lambda)^{-1} ) $ discounted regret for any $ \lambda $ in a defined continuous interval.

**Strengths:**

* The paper studies an important open problem: that of achieving discounted regret bounds with an unknown $ \lambda $. The paper does a decent job of motivating that in some settings $ \lambda $ is truly unknown and is not just a tunable hyperparameter.

* In contrast to typical aggregation mechanism in meta algorithms with multiple instances of an online gradient descent or experts algorithm, this work relies on the less well-known mechanism of discounted normal predictor with conservative updates. The authors show that this aggregator, works used in a particular order of $ \lambda $.

* The authors also provide some empirical results that reflect their theoretical findings.

* The authors rely on existing algorithms (e.g., aggregation with DNP-cu was already used in Zhang et al. (2022)). However, they prove an important property of DNP-cu: that it can aggregate across different discount factors.

**Weaknesses:**

* There is a related line of work on online convex optimization with unbounded memory (Kumar, Dean, Kleinberg, NeurIPS 2023). One special case is $\rho$-discounted infinite memory, where the loss in each round depends on the entire history of decisions and each past decision is weighted by a geometric factor of $\rho$. This paper does not discuss similarities and differences from this line of work.

* Algorithm 3 is hard to read - I had to keep jumping around to look at the algorithm and at the equations that it references. It would be easier for the reader if you wrote the equations inline.

* Since the algorithm is the same in Zhang et al. (2022) but you prove an additional property of DNP-cu that allows it to be used in the $\lambda$-discounted setting, can you discuss a bit more what are the differences between this work and Zhang et al.?

**Questions:**

* Please see the weaknesses section about lack of discussion on online convex optimization with unbounded memory. Could you discuss when using one model over the other is preferable? Can their techniques be extended to $\lambda$-discounted regret? Can your techniques be used to derive results for their model?

* Can you expand a bit more on why the ordering is crucial when aggregating the difference OGD instances?

---

> ### Author Response · Authors · 2025-11-20
>
> **Many thanks for the constructive reviews! We have revised our paper accordingly, including related work on OCO with memory, and Algorithm 3. Please refer to our revised version for details.**
>
> ---
>
> **Q1:** More discussions on related work of online convex optimization with unbounded memory.
>
> **A1:** We sincerely thank the reviewer for bringing this related line of work to our attention. Online convex optimization with unbounded memory allows the loss in the current round to depend on the entire history of decisions until that points, which can capture the complete long-term dependence of current losses on past decisions and model complex temporal-structure applications, such as online linear control.
>
> Both discounted OCO and OCO with unbounded memory extend the classical OCO framework by introducing dependence on past. Specifically, OCO with unbounded memory defines the policy regret as follows:
> $$
> \mathcal{R}\_T(\mathcal{A})=\sum_{t=1}^T f_t ( h_t ) - \min_{x\in \mathcal{X}}\sum_{t=1}^T f_t \left(\sum_{k=0}^{t-1}A^{k} B x\right)
> $$
> where they set $h_t = Ah_{t-1}+B x_t$ be the history of decisions, and $A$ and $B$ are operators. When considering the special case you mentioned, OCO with $\rho$-discounted infinite memory, they set $A(y_0,y_1,\cdots)=(0,\rho y_0,\rho y_1,\cdots)$, which implies the history will be
> $$
> h_t=(x_t,\rho x_{t-1},\rho^2 x_{t-2},\cdots )
> $$
> where $\rho$ is a discount factor. It can be seen that the action history in their formulation incorporates discounting, thereby placing greater emphasis on recent actions while diminishing the influence of actions taken further in the past.
>
> Compared to the definition of the discounted regret in our work, these two settings consider fundamentally different notions of how past losses are incorporated. In our discounted OCO formulation, the discount factor is applied directly to the final loss function, and related to the performance measurement. In contrast, their model incorporates the discount factor to the entire history of decisions, as shown in $h_t$, with the extent of historical influence determined by the size of the memory.
>
> We have included this line of related work in the revised related work section and provided a detailed discussion of its connections to and differences from our setting. Please refer to **Section 2.2 of related work in our revised version** for details.
>
> ---
>
> **Q2:** Could you discuss when using one model ... Can your techniques be used to derive results for their model?
>
> **A2:** We respond to the three questions you raised as follows:
>
> * **When using one model over the other is preferable?** The two models are based on different problem settings, as detailed in our response to Q1. Therefore, which model to use depends on the type of problem. For example, in certain economic models and online advertising scenarios, the discounted OCO framework is appropriate because the goal is to optimize the current decision while placing more emphasis on recent performance. In contrast, for embodied robotics and game agents, OCO with unbounded memory is more suitable, since past actions continue to influence future system behavior.
> * **Can their techniques be extended to discounted OCO?** No. Their techniques do not involve special treatment of the discount factor. As can be seen from the proof of Theorem C.2 in their work, they only use $\Vert A^k \Vert=\rho^k$ to bound $\tilde{L}$ and $H_p$, where $A(y_0,y_1,\cdots)=(0,\rho y_0,\rho y_1,\cdots)$.
> * **Can your techniques be used to derive results for their model?** No. In principle, one could construct multiple expert-algorithms with different learning rates corresponding to different discount factors. However, the main difficulty lies in the design of the meta-algorithm. The DNP-cu in our work, as well as other existing meta-algorithms, cannot be directly applied to the setting of OCO with unbounded memory.
>
> ---
>
> **Q3:** Algorithm 3 is hard to read. It would be easier for the reader if you wrote the equations inline.
>
> **A3:** Thank you for the helpful suggestion. We have revised our paper accordingly. Please refer to **Algorithm 3 in our revised version** for details.

---

> ### Author Response · Authors · 2025-11-20
>
> **Q4:** More discussions on the differences between this work with Zhang et al. (2022).
>
> **A4:** We are happy to provide further discussion. Since Zhang et al. (2022) focus on adaptive regret with switching costs, their analysis for DNP-cu is restricted to the standard payoff (Theorem 1, Zhang et al., 2022), where for any interval $[r,s]\subseteq [T]$ with the length $\tau$, DNP-cu satisfies
> $$
> \sum_{t=r}^{s} \left( g(x_t) b_t - \frac{1}{\mu} \left| g(x_t) - g(x_{t+1}) \right| \right)\geq \max \left( 0, \sum_{t=r}^{s} b_t - \frac{\tau}{n} \left( U(n) + 2\mu \right) - U(n) - \mu - Z\tau \right).
> $$
> The inequality above is sufficient for their analysis to aggregate the decisions of multiple experts, as the experts in their work operate on the *same* performance measures. However, under the setting of discounted OCO, the unknown discount factor causes the experts to operate under *different* measures. Therefore, we need to present the performance of DNP-cu under the discounted payoff setting.
>
> To achieve this, we provide a new theoretical guarantee (Theorem 2) to demonstrate that DNP-cu can support $\lambda$-discounted regret when the parameter $\rho=\lambda$ is set, and at the same time, it can provide a guarantee on the discounted payoff for a different discount factor, which proves the theoretical feasibility of the algorithm design. Compared to the standard payoff analysis, we first show that the discounted payoff with a different factor $\eta\geq\rho$ can be effectively bounded. To this end, we propose a new analytical approach to handle DNP-cu with a different parameter. Furthermore, we need to struggle with the potential function in Theorem 4. Bounding this potential function is particularly challenging, and previous work does not provide a rigorous proof. In our work, we establish a new lemma, i.e., Lemma 1, by adopting an insightful approach to provide an entirely new analysis to upper bound this function.
>
> ---
>
> **Q5:** Can you expand a bit more on why the ordering is crucial when aggregating the difference OGD instances?
>
> **A5:** We are happy to provide further discussion on the ordering. Our strategy for aggregating the expert decisions is illustrated in Figure 2 (Line 108 to 119), where we combine the decisions of multiple experts sequentially. The key theoretical guarantee of this framework comes from Theorem 2 (Line 355 to 369 in revised veriosn), which requires that $\eta \ge \rho$. Therefore, we construct the sequence of algorithms in a descending order of their associated discount factors, ensuring that the condition $\lambda_i\geq\lambda_{i+1}$ for all $i$ in range of the number of experts is satisfied when aggregating the decisions sequentially.

---

### Official Review · Reviewer_LMJ1 · 2025-10-31

**Soundness:** 3
**Presentation:** 2
**Contribution:** 2
**Rating:** 4
**Confidence:** 3

**Summary:**

The paper studies discounted online convex optimization and demonstrates that it is possible to achieve regret results for unknown bounded discount factors. It achieves this by analyzing and utilizing the existing algorithm DNP.

**Strengths:**

Originality:
The paper creatively applies the DNP algorithm to the discounted optimization problem and remove the need to know the discount factor.

Quality:
The submission seems technically correct. Experiments are a plus. The algorithms and the experimental settings appear reproducible.

Clarity:
The submission is clear in general.

Significance:
Theoretical novel findings in the form of discounted regret results for unknown discount factors are obtained.

**Weaknesses:**

I am leaning towards rejection. Below are the reasons.

Utilizing DNP-cu seems to help in arriving to a clean result, but it seems any combiner could have worked. I am not sure if the issue of different discounted performance measures as explained in Figure 1 is as great as advertised. Since the combined $\lambda$ values have a difference of $1/T$, the regret redundancy propagating due to mismatches seems to be finite at each combination node, similar to DNP-cu.

Aside from that, exponentially growing step-sizes and iterative combination of experts (with intermediate experts in the mix) are established methods in the literature.

**Questions:**

Questions:

Page 6 Line 301: is the effective window size the sum of the discount coefficients?

Page 6 Line 305: for $\lambda$, $\tau$ is related to the lower-bound, while the footnote is talking about the upper-bound. Is there a mistake here?

Page 8 Line 409: why $Z=1/T$, why not smaller?


Suggestions:

Give more substance to the need to use DNP as opposed to another mixture of experts algorithm.

---

> ### Author Response · Authors · 2025-11-20
>
> **Many thanks for the constructive reviews! We have revised our paper accordingly, including the typo in footnote. Furthermore, we would like to clarify a misunderstanding regarding our theoretical result. As detailed in our response to Q1, we have provided a thorough clarification. We would greatly appreciate it if you could kindly re-evaluate the contributions of our work.**
>
> ---
>
> **Q1:** Utilizing DNP-cu seems ... but it seems any combiner could have worked ... the regret redundancy propagating due to mismatches seems to be finite.
>
> **A1:** We believe there is a misunderstanding. **Our theoretical results cannot be obtained with any combiner**. For example, suppose we replace the DNP-cu combiner in Figure 2 with the well-established Hedge algorithm to aggregate the first two experts with their respective discount factors $\lambda_1=1-1/T,\lambda_2=1-2/T$. By the theoretical guarantee of Hedge, the meta-regret can be bounded by
> $$
> \sum_{t=1}^T f_t(\mathbf{w}\_t) - \sum_{t=1}^T f_t(\mathbf{w}\_{t,1}) \leq O(\sqrt{T \ln 2}),\quad \sum_{t=1}^T f_t(\mathbf{w}\_t) - \sum_{t=1}^T f_t(\mathbf{w}\_{t,2}) \leq O(\sqrt{T \ln 2}),
> $$
> where $\mathbf{w}\_{t,1}$ and $\mathbf{w}\_{t,2}$ denote the outputs from two experts. However, for expert-regret, we have
> $$
> \sum_{t=1}^T \lambda_1^{T-t} f_t(\mathbf{w}\_{t,1}) - \sum_{t=1}^T \lambda_1^{T-t} f_t(\mathbf{w}) \leq O\left(\frac{1}{\sqrt{1-\lambda_1}}\right),\quad \sum_{t=1}^T \lambda_2^{T-t} f_t(\mathbf{w}\_{t,2}) - \sum_{t=1}^T \lambda_2^{T-t} f_t(\mathbf{w}) \leq O\left(\frac{1}{\sqrt{1-\lambda_2}}\right).
> $$
> It is clear that combining the meta-regret with the expert-regret cannot yield any valid discounted regret guarantee, because the meta-algorithm and the expert-algorithm operate on different performance measurements.
>
> Furthermore, **the differences between the combined values are not** $1/T$. As our method uses exponentially increasing step sizes, $\lambda_i=1-2^i/T$, only the first two experts have discount factors that differ by $1/T$. The subsequent differences grow as $2/T$, $4/T$, and so on, up to $2^N/T$, where $N=\lceil \log_2 \frac{T}{\tau} \rceil$. **Therefore, the regret redundancy that propagates along the aggregation path is not finite**. This type of mismatch in the performance measurements cannot be addressed by the traditional meta-expert framework as discussed in Section 1.1.
>
> ---
>
> **Q2:** Exponentially growing step-sizes and iterative combination of experts (with intermediate experts in the mix) are established methods in the literature.
>
> **A2:** There do exist relevant works in the literature. However, all of these methods focus on *fixed* performance measurements. In our work, the discount factor is unknown, which causes the performance measurement to vary and leads to the technical challenge described in Section 1.1. To address this case, we provide a novel analysis revealing that DNP-cu is able to successfully aggregate the decisions of two experts, *even when they operate on discounted regret with different discount factors*.
>
> ---
>
> **Q3:** Is the effective window size the sum of the discount coefficients?
>
> **A3:** Yes, you are right. We treat the sum of the discount coefficients, i.e., $\sum_{t=1}^{T} \lambda^{T-t} =1/(1-\lambda)$ when $T\rightarrow \infty$, as the effective window size.
>
> ---
>
> **Q4:** For $\lambda,\tau$ is related to the lower-bound, while the footnote is talking about the upper-bound. Is there a mistake here?
>
> **A4:** Indeed, this is a typo. Thank you for pointing it out! We have corrected the footnote in the revised version, which is
>
> > While there is no particular reason to avoid setting the upper bound of $\lambda$ to $1-1/T^\alpha$ for some $\alpha \geq 1$, we focus on $1-1/T$ for simplicity.
>
> ---
>
> **Q5:** For $Z=1/T$, why not smaller?
>
> **A5:** As detailed in Lines 908 to 917, our final bound is given by
> $$
> \sum_{t=1}^T \lambda^{T-t} [f_t(\mathbf{w}_t)-f_t(\mathbf{w})] \leq \frac{2GD}{\sqrt{1-\lambda}} \left(4\sqrt{\log \textcolor{red}{\frac{1}{Z}}}+\sqrt{2}\right) + \frac{GD(N+1)\textcolor{red}{Z}}{1-\lambda}+2GD.
> $$
> From the above bound, it can be seen that the regret exhibits a trade-off with respect to $Z$. When we set $Z = 1/T$, the second term becomes $Z/(1-\lambda)=O(1)$, making it non–leading. Choosing a smaller $Z$ would worsen the dependence on the first term, thereby providing no benefit to the overall bound.
>
> ---
>
> **We hope that our responses can address your concerns, and we would greatly appreciate it if you could re-evaluate the contributions of our work.**

---

### Official Review · Reviewer_DsSC · 2025-11-01

**Soundness:** 3
**Presentation:** 3
**Contribution:** 3
**Rating:** 6
**Confidence:** 3

**Summary:**

The paper addresses online convex optimization (OCO) with discounted regret, where recent losses are weighted more heavily than distant ones via a discount factor $\lambda$. Existing papers assume the discounting factor is known, whereas this manuscript considers the unknown $\lambda$ case. The main idea is to run many instances of SOGD with a specific lambda, and then run a meta selection algorithm to aggregate the outcome.

**Strengths:**

Addresses a clear open problem in discounted OCO, by providing a discounted regret bound for uniform range of lambda. The learner does not know lambda a priori.

The technical challenges of the problem were clearly presented. The prior aggregation framework needed to operate under a uniform performance metric, whereas in this paper each expert has a different metric.

Provide a step by step derivation on the motivation behind the algorithm design which is insightful.

 Provides rigorous theoretical analysis and complete proofs. Simulations are provided as well.

**Weaknesses:**

-	I encourage the authors to provide **more concrete** motivations on the relevance of the problem of unknown lambda, why it is practically relevant other than the mere theoretic interest.
-	As mentioned in the intro, part of the motivation is that the user’s preference might change, indicating a time-varying lambda. Can the paper be extended to the time-varying lambda case? i.e. the learner has some feedback signal that is indicative of lambda, and can adapt itself to optimize the regret with time-varying lambda?

**Questions:**

-	(17) (18) are not too obvious, providing more details will help.

-	Compare theorem 3 to theorem 1, the dependence on lambda is worse. I wonder whether the bound is tight or if there are potential ways to further improve it.

---

> ### Author Response · Authors · 2025-11-20
>
> **Many thanks for the constructive reviews! We have revised our paper accordingly, including more concrete motivations, future work on time-varying case, and clearer equations. Please refer to our revised version for details.**
>
> ---
>
> **Q1:** Provide more concrete motivations on the relevance of the problem of unknown $\lambda$.
>
> **A1:** Thank you for the suggestion! Below we provide two practical scenarios where the discount factor $\lambda$ is unknown in practice, yet choosing an "true" value is crucial.
>
> * **Control Systems:** In many control systems, including LQR and other infinite-horizon formulations, discounting is introduced to place greater emphasis on near-term system behavior rather than distant future states. In realistic scenarios, this discount factor is typically determined based on system dynamics, time delays, or reliability decay. Therefore, this factor cannot be directly chosen by the designer, as doing so would lead to unstable controlling.
> * **Economic Decision Modeling:** In financial forecasting and portfolio optimization, the discount factor determines how rapidly past returns are devalued when assessing current market conditions.  Its value is implicitly determined by latent market characteristics such as long-term dependence, autocorrelation strength, and structural stability. Thus, the discount factor is not an arbitrary design choice but an unknown parameter rooted in real market behavior, requiring careful estimation through empirical studies or behavioral assumptions.
>
> In the revised version, we provide these concrete motivations based on practical scenarios. Please refer to **Appendix E Practical Motivations** for details.
>
> ---
>
> **Q2:** Can the paper be extended to the time-varying $\lambda$ case?
>
> **A2:** When the sequence of time-varying discount factors $\lambda_{1:T}=(\lambda_1,\lambda_2,\cdots,\lambda_T)$ is *known* in advance, Zhang et al. (2024) propose FTRL-based magnitude learner, which achieves $O(\sqrt{\sum_{t=1}^T (\sum_{i=1}^{t-1} \lambda_i^2)})$ discounted regret (Theorem 4, Zhang et al., 2024). However, when the time-varying sequence $\lambda_{1:T}$ is *unknown*, we found this to be highly challenging, as it is difficult to handle the time-varying case within the analysis of DNP-cu. Specifically, when the discount factor is unknown and varies over time, it becomes extremely difficult to control the discounted payoff in a manner analogous to Theorem 2. Naturally, the subsequent analysis no longer holds.
>
> We believe that addressing the case of the unknown time-varying $\lambda_{1:T}$ is a very meaningful direction for future research in discounted OCO, and we have included it in the discussion of future work. Please refer to **Section 5 Future Work in our revised version** for details. Thank you for raising this insightful question!

---

> ### Author Response · Authors · 2025-11-20
>
> **Q3:** Equations (17) (18) are not too obvious, providing more details.
>
> **A3:** We are happy to provide additional clarifications to make the equations (17) and (18) clearer. By the convexity of $f_t(\cdot)$, we have
> $$
> f_t(\mathbf{w}\_t)\leq (1-\omega_t)f_t(\mathbf{w}\_{t,1}) + \omega_t f_t(\mathbf{w}\_{t,2}),
> $$
> where the decision $\mathbf{w}\_t=(1-\omega_t)\mathbf{w}\_{t,1}+\omega_t\mathbf{w}\_{t,2}$ denotes the convex combination of $\mathbf{w}\_{t,1}$ and $\mathbf{w}\_{t,2}$ from each expert, and the weight $\omega_t\in [0,1]$. Then, we multiply both sides of the above inequality by $\lambda_1^{T-t}$ and take a cumulative summation to obtain
> $$
> \sum_{t=1}^T \lambda_1^{T-t} f_t(\mathbf{w}\_t)\leq \sum_{t=1}^T \lambda_1^{T-t}(1-\omega_t)f_t(\mathbf{w}\_{t,1}) + \sum_{t=1}^T \lambda_1^{T-t}\omega_t f_t(\mathbf{w}\_{t,2}).
> $$
> By rearranging the terms, Equation (17) is attained
> $$
> \sum_{t=1}^T \lambda_1^{T-t} f_t(\mathbf{w}\_t) - \sum_{t=1}^T \lambda_1^{T-t} f_t(\mathbf{w}\_{t,1}) \leq -\sum_{t=1}^T \lambda_1^{T-t}\omega_t (f_t(\mathbf{w}\_{t,1})-f_t(\mathbf{w}\_{t,2}))=-GD\sum_{t=1}^T \lambda_1^{T-t} \omega_t\ell_t,
> $$
> where the last equality is due to the definition of $\ell_t$, which is
> $$
> \ell_t = \frac{f_t(\mathbf{w}\_{t,1})-f_t(\mathbf{w}\_{t,2})}{GD}\in [-1,1].
> $$
> The derivation for Equation (18) follows in a similar manner. We start with the first inequality, and multiply both sides of the above inequality by $\lambda_2^{T-t}$ and take a cumulative summation to obtain
> $$
> \sum_{t=1}^T \lambda_2^{T-t} f_t(\mathbf{w}\_t)\leq \sum_{t=1}^T \lambda_2^{T-t}(1-\omega_t)f_t(\mathbf{w}\_{t,1}) + \sum_{t=1}^T \lambda_2^{T-t}\omega_t f_t(\mathbf{w}\_{t,2}).
> $$
> By rearranging the terms, Equation (18) is achieved
> $$
> \sum_{t=1}^T \lambda_2^{T-t} f_t(\mathbf{w}\_t) - \sum_{t=1}^T \lambda_2^{T-t} f_t(\mathbf{w}\_{t,1}) \leq -\sum_{t=1}^T \lambda_2^{T-t}(\omega_t-1) (f_t(\mathbf{w}\_{t,1})-f_t(\mathbf{w}\_{t,2}))=-GD \sum_{t=1}^T \lambda_2^{T-t} (\omega_t\ell_t -\ell_t),
> $$
> where the last equality is due to the definition of $\ell_t$.
>
> In the revised version, we have refined this part to make the equation derivations clearer. Please refer to **our revised version (Line 395 to 397)** for details.
>
> ---
>
> **Q4:** The dependence on $\lambda$ is worse in Theorem 3 compared to Theorem 1. Whether the bound is tight or if there are potential ways to further improve it.
>
> **A4:** We would like to clarify that the dependence on $\lambda$ has not worsened, as it remains governed by the same $O(1/\sqrt{1-\lambda})$ in both Theorem 1 and Theorem 3. It is possible that the second term in Theorem 3 led to some misunderstanding.
>
> According to the interval of $\lambda$ defined in our work (Line 315 to 317 in the revised version), we have $1/(1-\lambda)\in [\tau,T]$. Hence, the second term in Theorem 3 satisfies
> $$
> \frac{1}{(1-\lambda) T} = O(1),
> $$
> which is not the leading term and can therefore be ignored.

---

### Meta-Review · Area_Chair_MQTT · 2026-01-12

**Summary:**

1. The motivations on the relevance of the problem of unknown lambda;
2. The extension to the time-varying lambda case;
3. Utilizing DNP-cu seems to help in arriving to a clean result, but it seems any combiner could have worked;
4. Comparison with (Kumar, Dean, Kleinberg, NeurIPS 2023) and Zhang et al. (2022).

**Reviewer Concerns:**

Addressed concerns:

2. The authors show the challenge for controling the discounted payoff in a manner analogous to Theorem 2, which make this concern as a future work.
3. The authors provide detailed examplanation of why the theoretical results cannot be obtained with any combiner.
4. The comparisons in the rebuttal is sufficient.

Outstanding concerns:

1. The authors provide Control Systems and Economic Decision Modeling as two example to support the motivation, but I think such motivations are more suitable for time-varing parameters.

**Reviewer Scores:**

No

---

### Decision · Program_Chairs · 2026-01-26

Accept (Poster)